# ON THE WASSERSTEIN GEODESIC PRINCIPAL COMPONENT ANALYSIS OF PROBABILITY MEASURES

**Nina Vesseron**
CREST-ENSAE, IP Paris
{nina.vesseron}@ensae.fr

**Elsa Cazelles** *
CNRS, IRIT, Université de Toulouse
{elsa.cazelles}@irit.fr

**Alice Le Brigant** *
Université Paris 1 Panthéon Sorbonne
{alice.le-brigant}@univ-paris1.fr

**Thierry Klein**
ENAC, IMT, Université de Toulouse
{thierry.klein}@math.univ-toulouse.fr

## ABSTRACT

**On the Wasserstein Geodesic Principal Component Analysis of probability measures** This paper focuses on Geodesic Principal Component Analysis (GPCA) on a collection of probability distributions using the Otto-Wasserstein geometry. The goal is to identify geodesic curves in the space of probability measures that best capture the modes of variation of the underlying dataset. We first address the case of a collection of Gaussian distributions, and show how to lift the computations to the space of invertible linear maps. For the more general setting of absolutely continuous probability measures, we leverage a novel approach to parameterizing geodesics in Wasserstein space with neural networks. Finally, we compare to classical tangent PCA through various examples and provide illustrations on real-world datasets.

## 1 INTRODUCTION

In this paper, we are interested in computing the main modes of variation of a dataset of absolutely continuous (a.c.) probability measures supported in $\mathbb{R}^d$. For data points living in an arbitrary Hilbert space, the classical approach defined by Principal Component Analysis (PCA) consists in finding a sequence of nested affine subspaces on which the projected data retain a maximal part of the variance of the original dataset, or equivalently, yield best lower-dimensional approximations. When dealing with a set of a.c. probability distributions, a natural choice is to identify the probability measures with their probability density functions and to perform PCA on these using the $L^2$ Hilbert metric. Unfortunately, as highlighted in Cazelles et al. (2018), the components computed in this manner fail to capture the intrinsic structure of the dataset: the projections onto the components most likely result in non-positive and un-normalized functions. Using the Wasserstein metric $W_2$ instead has proven to overcome these limitations by taking into account the geometry of the space of distributions.

The Wasserstein metric endows the space of probability distributions with a *Riemannian-like* structure, framing the problem as PCA on a (positively) curved Riemannian manifold. A first approach to solve this task, known as *Tangent PCA* (TPCA), consists in embedding the data into the tangent space at a reference point, and applying classical PCA in this flat space, as in Fletcher et al. (2003). TPCA is computationally advantageous but can generically induce distortion in the embedded data, depending on the curvature of the manifold at the reference point and the dispersion of the data. A more geometrically coherent approach is *Geodesic PCA* (GPCA) proposed for Riemannian manifolds in Huckemann et al. (2010); Huckemann & Ziezold (2006), where principal modes of variations are geodesics that minimize the variance of the projection residuals. Following this approach, the first geodesic component of a set of probability measures $\nu_1, \ldots, \nu_n$ in the Wasserstein space solves

$$\inf_{t \mapsto \mu(t) \text{ geodesic}} \sum_{i=1}^{n} \inf_{t_i} W_2^2(\mu(t_i), \nu_i). \tag{1}$$

---

* These authors contributed equally to this work.

Interestingly, unlike in the Hilbert setting, this criterion is *not* equivalent to maximizing the variance of the projections, which leads to a different notion of PCA on Riemannian manifolds (see Sommer et al. (2010; 2014)).

**Related works**   TPCA in the Wasserstein space was considered by Wang et al. (2013) through the use of the linearized Wasserstein distance. In a similar approach, Boissard et al. (2015) restrict to distributions that can be obtained by deforming a single template measure. For one-dimensional probability measures, Bigot et al. (2017) have shown that GPCA and its linearized approximation TPCA coincide, as the embedding into a tangent space is then an isometry when constrained to a convex set. An algorithm in this case has been proposed in Cazelles et al. (2018), with an approximate extension in dimension 2. For higher-dimensional measures, Seguy & Cuturi (2015) solve an approximate version of GPCA, replacing geodesics by generalized geodesics as defined in Ambrosio et al. (2008). Despite all this, a method to solve the *exact* GPCA problem described in equation 1 is still missing for $\mathbb{R}^d$-valued probability measures. The goal of this paper is to fill this gap.

**Main contributions**   In this paper, we introduce two algorithms to solve the exact GPCA problem in the Wasserstein space of (1) centered Gaussian distributions and (2) a.c. probability measures on $\mathbb{R}^d$. Our methods are exact in the sense that they do not rely on a linearization of the Wasserstein space, and the components are true geodesics that minimize the cost in equation 1. In the Gaussian case, we leverage the Bures-Wasserstein geometry to lift the computations to the flat space of invertible matrices. We show an example where GPCA and TPCA differ significantly, and relate this effect to curvature. In the general case of a.c. probability distributions, we lift the probability distributions to the space of (non necessarily optimal) maps that pushforward a given reference measure, as described by Otto (2001). This approach is independent of the chosen reference measure and yields a convenient way to parametrize geodesic components and define orthogonality with respect to the Wasserstein metric. In practice, we parametrize geodesic components using multilayer perceptrons (MLPs), trained to minimize the cost in equation 1. We show illustrations on images and 3D point clouds. Along the way, we prove that for univariate Gaussian distributions, GPCA yields the same results whether it is performed in the space of a.c. distributions or restricted to the Gaussian submanifold.

**Organization of the paper**   In Section 2, we present the Wasserstein metric and its restriction to Gaussian distributions, as well as the related Bures-Wasserstein and Otto-Wasserstein geometries. We present GPCA for centered Gaussian distributions in Section 3, and the general case of a.c. probability measures in Section 4. Experiments are presented in Section 5, and the paper ends with a discussion in Section 6. All the proofs and additional experiments are deferred to the appendices.

## 2   BACKGROUND

**The Wasserstein distance**   Optimal transport is about finding the optimal way to transport mass from one distribution $\mu$ on $\mathbb{R}^d$ to another $\nu$ with respect to a ground cost, say the Euclidean squared distance. The total transport cost defines the *Wasserstein distance* $W_2$ between a.c. measures $\mu, \nu$ with moment of order 2, whose Monge (1781) formulation is given by

$$W_2^2(\mu, \nu) = \int_{\mathbb{R}^d} \|x - T_\mu^\nu(x)\|^2 d\mu(x), \tag{2}$$

and where the map $T_\mu^\nu$ is the $\mu$-a.s. unique gradient of a convex function verifying $T_\mu^\nu \# \mu = \nu$, as proven by Brenier (1991). When the distributions $\mu$ and $\nu$ are centered (non-degenerate) Gaussian distributions, they can be identified with their covariance matrices $\Sigma_\mu, \Sigma_\nu$ and the induced distance on the manifold $S_d^{++}$ of symmetric positive definite (SPD) matrices is called the *Bures-Wasserstein distance* $BW_2$ (see e.g. Modin (2017); Bhatia et al. (2019)):

$$BW_2^2(\Sigma_\mu, \Sigma_\nu) = \text{tr} \left[ \Sigma_\mu + \Sigma_\nu - 2(\Sigma_\mu^{1/2} \Sigma_\nu \Sigma_\mu^{1/2})^{1/2} \right]. \tag{3}$$

Both distances can be induced by a Riemannian metric on their respective manifolds, i.e. the space of a.c. distributions and $S_d^{++}$, as we will see in the following. For more details, see Appendix B.

**Bures-Wasserstein geometry of centered Gaussian distributions**   The set of centered non-degenerate Gaussian distributions on $\mathbb{R}^d$ is identified with the manifold $S_d^{++}$ of SPD matrices.

The Riemannian geometry of the Bures-Wasserstein metric in equation 3 can be described by considering $S_d^{++}$ as the quotient of the manifold $GL_d$ of invertible matrices by the right action of the orthogonal group $O_d$. In this geometry, $GL_d$ is decomposed into equivalence classes called *fibers*. The fiber over $\Sigma \in S_d^{++}$ is defined to be the pre-image of $\Sigma$ under the projection

$$\pi\colon A \in GL_d \mapsto AA^\top \in S_d^{++}, \tag{4}$$

and can be obtained as the result of the action of $O_d$ on a representative, e.g. $\Sigma^{1/2}$ the only SPD square root of $\Sigma$: $\pi^{-1}(\Sigma) = \{A \in GL_d, \ AA^\top = \Sigma\} = \Sigma^{1/2} O_d$.

Tangent vectors to $GL_d$ are said to be *horizontal* if they are orthogonal to the fibers with respect to the Frobenius metric, i.e. if they belong to the space

$$\mathrm{Hor}_A\colon \ = \{X \in \mathbb{R}^{d \times d}, \ X^\top A - A^\top X = 0\}, \quad (5)$$

for a given point $A \in GL_d$. Then the projection $\pi$ in equation 4 defines an isometry between the horizontal subspace $\mathrm{Hor}_A$ equipped with the Frobenius inner product $\langle X, Y \rangle\colon \ = \mathrm{tr}(XY^\top)$, and $S_d^{++}$ equipped with a Riemannian metric that induces the Bures-Wasserstein distance (equation 3) as the geodesic distance. In particular, this means that moving horizontally along straight lines in the top space $GL_d$ is equivalent to moving along geodesics in the bottom space $S_d^{++}$ (see Figure 1), as recalled in the following proposition.

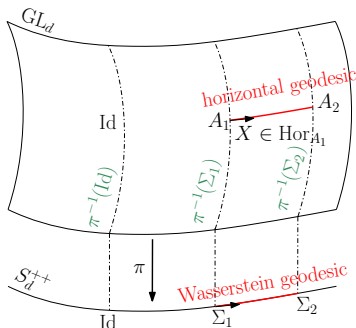

Figure 1: The Bures-Wasserstein geometry of centered Gaussian distributions, inspired by Khesin et al. (2021).

**Proposition 1** (Takatsu (2011); Malagò et al. (2018); Bhatia et al. (2019)). *Any geodesic $t \mapsto \Sigma(t)$ in $S_d^{++}$ for the Bures-Wasserstein metric in equation 3 is the $\pi$-projection of a horizontal line segment in $GL_d$, that is*

$$\Sigma(t) = \pi(A + tX) = (A + tX)(A + tX)^\top, \quad A \in GL_d, \ X \in \mathrm{Hor}_A, \tag{6}$$

*where $t$ is defined in a certain time interval $(t_{min}, t_{max})$. Also, the Bures-Wasserstein distance between two covariance matrices $\Sigma_1, \Sigma_2 \in S_d^{++}$ is given by the minimal Euclidean distance between their fibers*

$$BW_2(\Sigma_1, \Sigma_2) = \inf_{Q_1, Q_2 \in O_d} \|\Sigma_1^{1/2} Q_1 - \Sigma_2^{1/2} Q_2\| = \inf_{Q \in SO_d} \|\Sigma_1^{1/2} - \Sigma_2^{1/2} Q\|, \tag{7}$$

*where $\|\cdot\|$ is the Frobenius norm and $SO_d$ is the special orthogonal group.*

It is essential to note that the geodesic equation 6 cannot be extended for all time $t \in \mathbb{R}$ (the only geodesic lines are those obtained by translation (Kloeckner, 2010, Proposition 3.6)). Therefore, equation 6 is only defined on a time interval $(t_{\min}, t_{\max})$ that depends on the eigenvalues of $XA^{-1}$ (see Appendix B.3). More details on the Bures-Wasserstein geometry can be found in Appendix B.2.

**Otto-Wasserstein geometry of a.c. probability measures** The Riemannian structure described for Gaussian distributions is a special case of Otto (2001)'s more general construction : the bottom space becomes the space $\mathrm{Prob}(\Omega)$ of a.c. distributions supported on a compact set $\Omega \subset \mathbb{R}^d$ while the top space is the space of diffeomorphisms $\mathrm{Diff}(\Omega)$ endowed with the $L^2$ metric with respect to a fixed reference measure $\rho$ (see Figure 19 in Appendix B). The *fibers* of $\mathrm{Diff}(\Omega)$ are then defined to be the pre-images under the projection

$$\pi\colon \varphi \in \mathrm{Diff}(\Omega) \mapsto \pi(\varphi) = \varphi_\# \rho \in \mathrm{Prob}(\Omega). \tag{8}$$

In this setting, *horizontal* displacements in $\mathrm{Diff}(\Omega)$ are along vector fields that are gradients of functions. The projection $\pi$ defines an isometry between the horizontal subspace equipped with the $L^2(\rho)$-inner product and $\mathrm{Prob}(\Omega)$ equipped with a Riemannian metric that induces the Wasserstein distance as the geodesic distance. In particular, we have the following result.

**Proposition 2** (Otto (2001)). *Any geodesic $t \mapsto \mu(t)$ for the Wasserstein metric given in equation 2 is the $\pi$-projection of a line segment in $\mathrm{Diff}(\Omega)$ going through a diffeomorphism $\varphi$ at horizontal speed $\nabla f \circ \varphi$ for some smooth function $f \in \mathcal{C}(\mathbb{R}^d)$. That is, for $t$ defined in a certain interval $(t_{min}, t_{max})$,*

$$\mu(t) = \pi(\varphi + t\nabla f \circ \varphi) = (\mathrm{id} + t\nabla f)_\#(\varphi_\# \rho). \tag{9}$$

*Another geodesic $\tilde{\mu}(t) = \pi(\varphi + t\nabla\tilde{f} \circ \varphi)$ is orthogonal to $\mu(t)$ at $t = 0$ for the Riemannian metric inducing the Wasserstein distance if and only if $\langle\nabla f \circ \varphi, \nabla\tilde{f} \circ \varphi\rangle_{L^2(\rho)} = 0$.*

We emphasize that $f$ need not be convex in equation 9, unlike in the more classical parametrization of geodesics due to McCann (1997) between two distributions $\mu_0$ and $\mu_1 = \nabla u_{\#}\mu_0$ :

$$\mu(t) = (\mathrm{id} + t(\nabla u - \mathrm{id}))_{\#}\mu_0, \text{ with } t \in [0,1] \text{ and } u \text{ a convex function.} \tag{10}$$

Note that equation 9 parametrizes geodesics provided that $\mathrm{id} + t\nabla f$ is a diffeomorphism, and thus it is defined on a time interval that depends on the eigenvalues of the Hessian of $f$. On the other hand, the convexity condition on the function $u$ in the parametrization of equation 10 ensures that time $t$ is defined on $[0,1]$. Both are completely equivalent (see Appendix B.3 for details).

## 3   GEODESIC PCA ON CENTERED GAUSSIAN DISTRIBUTIONS

In this section, we consider the exact GPCA problem for the Bures-Wasserstein metric in equation 3. The data are $n$ centered Gaussian distributions identified with their covariance matrices $\Sigma_1, \ldots, \Sigma_n \in S_d^{++}$. Following Huckemann et al. (2010), we define the first component as the geodesic $t \mapsto \Sigma(t) \in S_d^{++}$ that minimizes the sum of squared residuals of the $BW_2$-projections of the data:

$$\inf_{t \mapsto \Sigma(t) \text{ geodesic}} \sum_{i=1}^{n} \inf_{t_i} BW_2^2(\Sigma(t_i), \Sigma_i). \tag{11}$$

The second principal component is defined to be the geodesic that minimizes the same cost function, with the constraint of intersecting the previous component orthogonally. The subsequent principal components have the additional constraint of going through the intersection of the first two principal geodesics. This definition does not impose that the geodesic components go through the Wasserstein barycenter (see Agueh & Carlier (2011)), and in Section 5 we show an example where this is indeed not verified. This gives an observation of the phenomenon already described in Huckemann & Ziezold (2006) for spherical geometry. The proofs of this section are deferred to Appendix D.

**Learning the geodesic components**   Following Propositon 1, we lift the GPCA problem in equation 11 to the total space $GL_d$ of Otto's fiber bundle. This has several advantages: the Bures-Wasserstein distance in the cost function of equation 11 is replaced by the Frobenius norm $\|\cdot\|$, the geodesic is replaced by a horizontal line segment, and the projection times $t_i$ become explicit. The price to pay is an optimization over variables $(Q_i)_{i=1}^n$ in $SO_d$, needed to represent the covariance matrices $\Sigma_i$ by invertible matrices $\Sigma_i^{1/2}Q_i$ in their respective fibers.

**Proposition 3.** *Let $\pi\colon GL_d \to S_d^{++}$, $A \mapsto AA^\top$ and $(A_1, X_1, (Q_i)_{i=1}^n)$ be a solution of*

$$\inf\ F(A_1, X_1, (Q_i)_{i=1}^n)\colon = \sum_{i=1}^{n} \|A_1 + p_{A_1,X_1}(t_i)X_1 - \Sigma_i^{1/2}Q_i\|^2, \tag{12}$$

$$\text{subject to}\quad A_1 \in GL_d,\ X_1 \in \mathrm{Hor}_{A_1},\ \|X_1\|^2 = 1,\ Q_1, \ldots, Q_n \in SO_d.$$

*Then there exist $t_{min}, t_{max} \in \mathbb{R}$ such that the geodesic $\Sigma \colon t \in [t_{min}, t_{max}] \mapsto \pi(A_1 + tX_1)$ in $S_d^{++}$ minimizes equation 11.*

Here the $t_i$ are projection times given by $t_i = \langle\Sigma_i^{1/2}Q_i - A_1, X_1\rangle$, and $p_{A,X}$ is a projection operator that clips any $t \in \mathbb{R}$ onto a closed interval $[t_{\min}, t_{\max}]$ depending on $A$ and $X$, such that $A + p_{A,X}(t)X$ is invertible for any $t$ in this interval (see Appendix B.3). Clipping the time parameter of the line segment is necessary to ensure it remains within $GL_d$ and projects onto a geodesic in $S_d^{++}$.

The second component is a geodesic of $S_d^{++}$ that orthogonally intersects the first component. Lifting again the problem to $GL_d$, this boils down to searching for a horizontal line $t \mapsto A_2 + tX_2$ where $A_2 = (A_1 + t^*X_1)R^*$ for a rotation matrix $R^*$, a time $t^* \in [t_{\min}, t_{\max}]$ and a horizontal vector $X_2 \in \mathrm{Hor}_{A_2}$ such that $\langle X_2, X_1R^*\rangle = 0$. The equation for $A_2$ ensures that the $\pi$-projections of the first two horizontal lines intersect, while the condition on $X_2$ ensures that they intersect orthogonally (since $X_1R^*$ is horizontal at $A_2$). See Figure 2. Since different choices of $R^*$ yield the same projected component in $S_d^{++}$, we fix $R^* = I_d$ in our algorithm to remove this ambiguity.

The second component is thus defined by $\Sigma_2(t) = \pi(A_2 + tX_2)$, found by solving:

$$\inf \ F(A_2, X_2, (Q_i)_{i=1}^n)$$

subject to $\quad A_2 = A_1 + t^* X_1,$

$t^* \in [t_{\min}, t_{\max}],\ X_2 \in \mathrm{Hor}_{A_2},\ \|X_2\|^2 = 1,$

$\langle X_2, X_1 \rangle = 0,\ Q_1, \ldots, Q_n \in SO_d.$

$$(13)$$

Note that this step requires to find new rotation matrices $(Q_i)_{i=1}^n$. The first two components fix the intersection point $\pi(A_2)$ through which all other geodesic components will pass, see Figure 2.

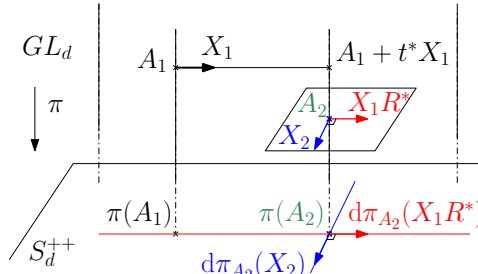

Figure 2: First (red) and second (blue) geodesic components of Gaussian GPCA, where $\mathrm{d}\pi_A$ denotes the differential of the projection $\pi \colon A \mapsto AA^\top$ at $A \in GL_d$.

For every higher order component, we search for a velocity vector $X_k$ that is horizontal at some point in the fiber over $\pi(A_2)$ and orthogonal to the lifts of the velocity vectors of the previous components. Details on the implementation of these components are given in Appendix D.2.

**Quantifying the difference between TPCA and GPCA**  In the following, we quantify the distortion induced by linearization in the case of covariances matrices with same eigenvalues.

**Proposition 4.** *Let $\Sigma \in S_2^{++}$ with eigenvalues $a^2, b^2$ and $\Sigma' = P_\theta \Sigma P_\theta^\top$ where $P_\theta$ is the rotation matrix of angle $\theta \neq 0 \pmod{2\pi}$. Then, denoting $\bar{\Sigma} = ((a+b)/2)^2 I$, we have*

$$\frac{BW_2^2(\Sigma, \Sigma')}{BW_{2,\bar{\Sigma}}^2(\Sigma, \Sigma')} = 1 - \left(\frac{a-b}{a+b}\right)^2 \cos^2 \theta + O((a-b)^4), \tag{14}$$

*where $BW_{2,\bar{\Sigma}}$ is the linearized Bures-Wasserstein distance at $\bar{\Sigma}$ recalled in equation 29.*

For a given $\theta$, equation 14 shows that the distorsion is most important for $\frac{|a-b|}{|a+b|}$ close to 1, which corresponds to matrices that are close to the border of the cone, as illustrated in Section 5.1.

**On the restriction to the space of Gaussian distributions**  Geodesic PCA can also be defined in the more general space of a.c. probability distributions, as presented in Section 4. A natural question that arises is whether performing GPCA in the whole space of probability distributions gives the same result as restricting to the space of Gaussian distributions, which is totally geodesic. The answer is yes in dimension one, as shown in Appendix D.

**Proposition 5.** *Let $\nu_i = \mathcal{N}(m_i, \sigma_i^2)$ for $i = 1, \ldots, n$, be $n$ univariate Gaussian distributions. The first principal geodesic component $t \in [0, 1] \mapsto \mu(t)$ solving equation 1 remains in the space of Gaussian distributions for all $t \in [0, 1]$.*

Up to our knowledge, this remains an open question in higher dimension.

# 4 GEODESIC PCA ON A.C. PROBABILITY MEASURES: GPCAGEN

We now tackle the task of performing GPCA on a set of a.c. probability measures $\nu_1, \ldots, \nu_n$ using the Otto-Wasserstein geometry. We propose a parameterization of the geodesic principal components based on Otto's formulation, leveraging neural networks. Additionally, we introduce a dedicated cost function to optimize the different geodesic components.

**Parameterizing geodesics**  Following Proposition 2 and equation 9, any geodesic $t \mapsto \mu(t)$ in the Wasserstein space $(\mathrm{Prob}(\Omega), W_2)$ can be expressed as $\mu(t) = (\varphi + t\nabla f \circ \varphi)_\# \rho$, for $t$ in some interval $[t_{\min}, t_{\max}]$, $\varphi \colon \mathbb{R}^d \to \mathbb{R}^d$ a diffeomorphism, $f \colon \mathbb{R}^d \to \mathbb{R}$ a smooth function, and $\rho$ a fixed reference measure, taken to be the standard Gaussian distribution in this work. Using multilayer perceptrons (MLPs) to parametrize the functions $\varphi$ and $f$, denoted $\varphi_\theta$ and $f_\psi$, respectively, the curve

$$t \mapsto \mu_{\theta,\psi}(t) = (\mathrm{id} + t\nabla f_\psi)_\# (\varphi_{\theta\#}\rho)$$

is a geodesic for $t \in [t_{\min}, t_{\max}]$, provided that $\mathrm{id} + t\nabla f_\psi \in \mathrm{Diff}(\Omega)$ for all $t$ in this interval. Equivalently, this condition holds if the Hessian matrix $I_d + tH_{f_\psi}(x)$ is positive definite for all $x \in \mathbb{R}^d$ and $t \in [t_{\min}, t_{\max}]$, where $H_{f_\psi}(x)$ denotes the Hessian of $f_\psi$ at $x$. In practice, we enforce this constraint by monitoring the eigenvalues of $I_d + tH_{f_\psi}(x)$ (see Appendix B.3) and either clipping $t$ or adjusting the interval $[t_{\min}, t_{\max}]$ to ensure that all eigenvalues remain positive. This representation enables to sample from the distributions along the geodesic. Specifically, given the learned vector field $\varphi_\theta$ and function $f_\psi$, one can sample from $\mu_{\theta,\psi}(t)$ by first drawing $x \sim \rho$ and then applying the transformations $\varphi_\theta$ and $\mathrm{id} + t\nabla f_\psi$ sequentially as $\varphi_\theta(x) + t\nabla f_\psi(\varphi_\theta(x)) \sim \mu_{\theta,\psi}(t)$.

**Learning the geodesic components**  The first principal component in GPCA minimizes the objective in equation 1. The scalar variables $t_i$ specify the projection time of each distribution $\nu_i$ onto the geodesic $t \mapsto \mu(t)$. Leveraging the explicit form of Otto's geodesic, equation 1 can be reformulated as:

$$\inf_{\substack{f \in \mathcal{C}(\mathbb{R}^d), \varphi \in \mathrm{Diff}(\Omega) \\ t_1, \ldots, t_n \in [t_{\min}, t_{\max}]}} \mathcal{L}(f, \varphi, t_1, \ldots, t_n) := \sum_{i=1}^n W_2^2((\mathrm{id} + t_i \nabla f)_\#(\varphi_\# \rho), \nu_i). \tag{15}$$

We jointly learn the parameters $t_i$ together with the neural networks $\varphi_\theta$ and $f_\psi$ to minimize the objective in equation 15. In practice, we use the Sinkhorn divergence $S_\varepsilon$ that has been proven to be a differentiable and computationally efficient approximation of the squared Wasserstein distance $W_2^2$, see Frogner et al. (2015); Genevay et al. (2018); Chizat et al. (2020), and represent the distributions $\rho$ and $\nu_i$ using batches of $m$ samples $x_k \sim \rho$ and $y_j \sim \nu_i$. The optimization proceeds by updating the parameters based on a single distribution $\nu_i$ sampled at each iteration, as detailed in Algorithm 1. To compute $t_{\min}$ and $t_{\max}$ on line 5 of Algorithm 1, we approximate the extremal eigenvalues of $H_{f_\psi}$ by evaluating the largest and smallest eigenvalues over the finite set $\{H_{f_\psi}(x_k)\}_{k=1}^m$, and substitute these estimates into the theoretical bounds from Appendix B.3.

---

**Algorithm 1** Geodesic PCA algorithm for a.c. measures: GPCAGEN

1: Initialize $\varphi_\theta$, $f_\psi$ and the $t_i$ for $1 \leq i \leq n$
2: **while** not converged **do**
3:   **for** $i = 1$ to $n$ **do**
4:     Draw $m$ i.i.d samples $y_j^{(i)} \sim \nu_i$ and draw $m$ i.i.d samples $x_k \sim \rho$   $1 \leq j, k \leq m$
5:     Estimate $t_{\min}, t_{\max}$ with $\{H_{f_\psi}(x_k)\}_{k=1}^m$ and set $t_i' = \min(\max(t_i, t_{\min}), t_{\max})$
6:     $z_k^{(i)} \leftarrow (\mathrm{id} + t_i' \nabla f_\psi) \circ (\varphi_\theta)(x_k)$  for $1 \leq k \leq m$
7:     $\mathcal{L}_{\theta,\psi,t_i} \leftarrow S_\varepsilon \left( \frac{1}{m} \sum_{k=1}^m \delta_{z_k^{(i)}}, \frac{1}{m} \sum_{j=1}^m \delta_{y_j^{(i)}} \right)$
8:     Update $\varphi_\theta$, $f_\psi$ and the $t_i$ with $\nabla \mathcal{L}_{\theta,\psi,t_i}$
9:   **end for**
10: **end while**

---

The second principal component minimizes the objective in equation 1 subject to the constraint that it intersects the first component orthogonally. Similar to the first component, we use two MLPs, $f_{\psi_2}$ and $\varphi_{\theta_2}$, to parameterize the geodesic $t \mapsto \mu_{\theta_2,\psi_2}(t)$, along with $n$ scalar variables $t_i^2$, to optimize the objective in equation 15. We also introduce two additional scalar variables, $t_{\mathrm{inter}}^1$ and $t_{\mathrm{inter}}^2$, which define the intersection times of the two geodesics, along with the regularization terms:

$$\mathcal{I}(\xi_1, \xi_2, t_{\mathrm{inter}}^1, t_{\mathrm{inter}}^2) = \|\xi_1(t_{\mathrm{inter}}^1) - \xi_2(t_{\mathrm{inter}}^2)\|_2^2 \quad \text{and} \quad \mathcal{O}(g, h) = \frac{\langle g, h \rangle_{L^2(\rho)}^2}{\|g\|_{L^2(\rho)}^2 \|h\|_{L^2(\rho)}^2},$$

where $\mathcal{I}$ enforces the two geodesics in $\mathrm{Diff}(\Omega)$, $\xi_1(t) = (\mathrm{id} + t\nabla f_\psi) \circ \varphi_\theta$ and $\xi_2(t) = (\mathrm{id} + t\nabla f_{\psi_2}) \circ \varphi_{\theta_2}$, to intersect at the respective times $t_{\mathrm{inter}}^1$ and $t_{\mathrm{inter}}^2$ while $\mathcal{O}(g, h)$ ensures orthogonality between the corresponding horizontal vector fields $g = \nabla f_\psi(\varphi_\theta)$ and $h = \nabla f_{\psi_2}(\varphi_{\theta_2})$ in $L^2(\rho)$. The total objective used to optimize the second principal component incorporates these regularization terms and is given by:

$$\mathcal{L}(f_{\psi_2}, \varphi_{\theta_2}, t_1^2, \ldots, t_n^2) + \lambda_I \mathcal{I}(\xi_{\theta,\psi}, \xi_{\theta_2,\psi_2}, t_{\mathrm{inter}}^1, t_{\mathrm{inter}}^2) + \lambda_O \mathcal{O}(\nabla f_\psi(\varphi_\theta), \nabla f_{\psi_2}(\varphi_{\theta_2}))$$

with $\xi_{\theta,\psi}(t) = (\mathrm{id} + t\nabla f_\psi) \circ \varphi_\theta$ and $\xi_{\theta_2,\psi_2}(t) = (\mathrm{id} + t\nabla f_{\psi_2}) \circ \varphi_{\theta_2}$ and where $\lambda_I$ and $\lambda_O$ are the regularization parameters controlling the trade-off between the intersection and orthogonality

regularization terms, respectively. Note that in virtue of Proposition 2, the $L^2(\rho)$ inner product in the regularization term $\mathcal{O}$ truly enforces orthogonality of the geodesic components with respect to the Riemannian metric associated to the Wasserstein distance. Note also that $\mathcal{I}$ enforces the geodesics to intersect in $\mathrm{Diff}(\Omega)$ which means that, at the intersection time, the geodesics $\mu_{\theta_1,\psi_1}$ and $\mu_{\theta_2,\psi_2}$ in $\mathrm{Prob}(\Omega)$ intersect and share the same representative. An alternative implementation would be to enforce the intersection of the geodesics $\mu_{\theta_1,\psi_1}$ and $\mu_{\theta_2,\psi_2}$ in $\mathrm{Prob}(\Omega)$ and to impose the orthogonality of $\nabla f_\psi(\varphi_\theta) \circ R^*$ and $\nabla f_{\psi_2}(\varphi_{\theta_2})$ in $L^2(\rho)$, where $R^* = \xi_{\theta_2,\psi_2}(t^2_{\mathrm{inter}}) \circ \xi_{\theta_1,\psi_1}(t^1_{\mathrm{inter}})^{-1}$. This approach is the one used in the Gaussian case. However, computing $R^*$ is computationally expensive, and we therefore preferred to impose $\xi_{\theta_1,\psi_1}(t^1_{\mathrm{inter}}) = \xi_{\theta_2,\psi_2}(t^2_{\mathrm{inter}})$ which directly yields $R^* = \mathrm{id}$.

The training algorithm used to optimize the second principal component follows the same structure as Algorithm 1, except for the seventh line, where the regularization terms, estimated using the minibatch $x_k \sim \rho$, are added to the loss function. Higher-order components can be computed similarly.

## 5 EXPERIMENTS

### 5.1 EXPERIMENTS ON CENTERED GAUSSIAN DISTRIBUTIONS

In this section, we consider toy examples in $S_2^{++}$ and compare GPCA to its widely used linearized approximation, TPCA (see Appendix C). We use two coordinate systems for matrices in $S_2^{++}$: the first comes from the spectral decomposition, and the second maps any SPD matrix to a point in the interior of the cone $\mathcal{C} = \{(x, y, z) \in \mathbb{R}^3, \; z > 0, \; z^2 > x^2 + y^2\}$:

$$\Sigma = P_\theta \begin{pmatrix} a^2 & 0 \\ 0 & b^2 \end{pmatrix} P_\theta^\top = \begin{pmatrix} z+y & x \\ x & z-y \end{pmatrix}, \quad (a, b, \theta) \in \mathbb{R}_+^* \times \mathbb{R}_+^* \times \mathbb{R}, \quad (x, y, z) \in \mathcal{C}, \quad (16)$$

where $P_\theta$ is the rotation matrix of angle $\theta$. Generically, GPCA and TPCA yield very similar results: for sets of $n = 50$ covariance matrices randomly generated using a uniform distribution on the parameters $(a, b, \theta)$, GPCA reduces the objective in equation 11 of less than $1\%$ w.r.t. TPCA, on average for 100 trials. This suggests that TPCA is generally a very good approximation of GPCA. Two extreme cases are described below: (i) GPCA and TPCA are equivalent and (ii) GPCA and TPCA drastically differ.

**Matrices with same orientation**  If we consider a set of covariance matrices that live in the subspace $\theta =$ constant in notations of equation 16, then both GPCA and TPCA yield exactly the same results,

namely that of linear PCA in the $(a, b)$-coordinates. This is because any such subspace has zero curvature for the Wasserstein metric, and geodesics are straight lines in the $(a, b)$-coordinates (Appendix D.1). Figure 3 shows the geodesic components obtained for a set of matrices in the subspace $\theta = 0$ that form a regular rectangular grid in the $(a, b)$ coordinates, i.e. $\Sigma_{ij} = \mathrm{diag}(a_i^2, b_j^2)$ where the $a_i$'s and $b_j$'s are equally spaced. They are indeed straight lines that capture the variations in $a$ and $b$ respectively.

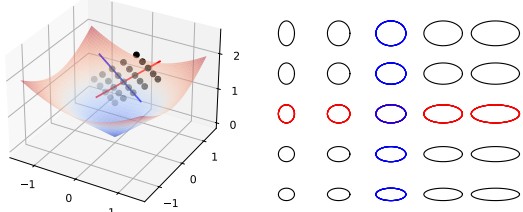

Figure 3: GPCA on a set of diagonal covariance matrices $\Sigma_{ij}$ with varying eigenvalues $1 \leq a_i^2 \leq 3$, $1 \leq b_j^2 \leq 2$. The matrices form a planar grid inside the cone $\mathcal{C}$ of SPD matrices in equation 16 (**left**), and correspond to ellipses of varying width and height (**right**). The first component (red) captures the variation in $a$, while the second component (blue) captures the variation in $b$.

**Matrices with same eigenvalues**  Now we consider covariance matrices that all have the same eigenvalues but different orientations. Specifically, we choose $\Sigma_i = P_{\theta_i}\mathrm{diag}(a^2, b^2)P_{\theta_i}^\top$, for positive reals $a > b$, $\theta_i = i\pi/n$ for $i = 0, \ldots, n-1$ and an even number $n$. In the $(x, y, z)$ coordinates (equation 16), the covariance matrices are displayed on a circle of equation $z = \mathrm{cst}$ (constant trace) and $x^2 + y^2 = \mathrm{cst}$ (constant determinant), as shown in Figure 4 (in practice, we choose a slightly open circle to break the symmetry). Then the Bures-Wasserstein barycenter of the covariance matrices

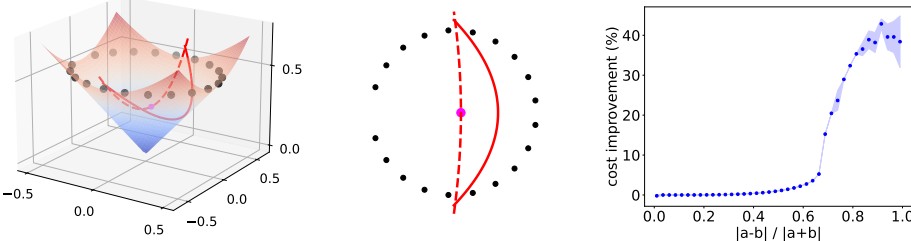

Figure 4: Comparison between tangent and geodesic PCA on a set of $n = 20$ covariance matrices with same eigenvalues $a^2, b^2$ and different orientations $\theta$. **(left)** They are equally spaced on an (open) circle in a horizontal plane inside the cone of SPD matrices. The first component of TPCA (dashed red line) goes through the Fréchet mean $\bar{\Sigma}$ (magenta dot), a multiple of the identity, while the component of GPCA (solid red line) does not. Here $|a - b|/|a + b| \approx 0.8$. **(middle)** Representation of the left figure in the $(x, y)$ coordinates. **(right)** Evolution of the first component cost improvement (in the sense of minimization) of GPCA with respect to TPCA, as a function of the ratio $|a - b|/|a + b|$.

$\Sigma_1, \ldots, \Sigma_n$ is given by $\bar{\Sigma} = (a + b)^2/4\,I$ (see Proposition 15 in Appendix D.1). When performing TPCA on $\Sigma_1, \ldots, \Sigma_n$ at the barycenter $\bar{\Sigma}$, the radial distances between $\bar{\Sigma}$ and $\Sigma_i$ are preserved, but not the pairwise distances between the $\Sigma_i$'s. Proposition 4 evaluates the level of this distorsion. Note that since $(a - b)^2/(a + b)^2 = (x^2 + y^2)/z^2$, the distorsion is most important when covariance matrices are close to the border of the cone, see Figure 4 **(left)**. Indeed, in that case, the results of GPCA can be very different from those of TPCA and the first component may not even go through the Wasserstein barycenter $\bar{\Sigma}$, see Figure 4 **(middle)** and Figure 8 in Appendix A. In that case GPCA may be seen as worse-behaved as TPCA, as some of the Gaussian distributions will project onto the first geodesic component boundaries, yielding a poor separation. Figure 4 **(right)** shows the percentage of improvement of the cost in equation 11 (in terms of minimization) of GPCA with respect to TPCA, in the setting previously described for different values of the ratio $|a - b|/|a + b|$. on average for 10 runs per value of the ratio. The blue strip indicates standard deviation.

**Weather dataset** In this paragraph, we use the Weather CORGIS Dataset to illustrate GPCA based on empirical covariance matrices. The dataset provides weekly measures of precipitation and wind speed recorded from March to January 2016 across the 50 U.S. states and the territory of Puerto Rico. From these measures, we construct two histograms for each state: one for precipitation and one for wind speed. We then compute the 50 empirical covariances from these histograms. We show in Figure 14 the projection of each state onto the two first GPCA components computed from the empirical covariance matrices. We can clearly identify clusters of different weather behavior among the states.

## 5.2 EXPERIMENTS ON ABSOLUTELY CONTINUOUS DISTRIBUTIONS

We conduct a preliminary experiment on a synthetic dataset with known geodesics to verify that our algorithm, GPCAGEN (Section 4), accurately recovers the first two principal components. We then apply GPCAGEN to 3D point clouds from the ModelNet40 dataset (Wu et al. (2015)) and to color distributions of images from the Landscape Pictures dataset (Rougetet (2020)). An additional

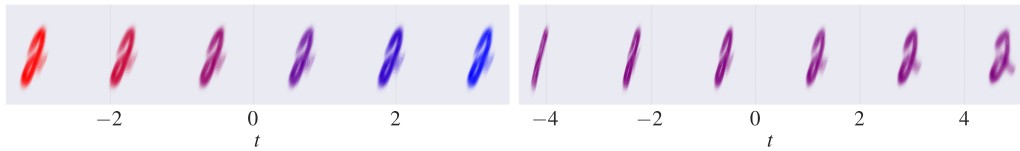

Figure 5: Densities of probability distributions uniformly sampled along the first and second principal geodesics components. GPCAGEN successfully recovers the two orthogonally intersecting geodesics constructed from MNIST data. The first component **(left)** captures variation in color space, while the second component **(right)** recovers the interpolation from the digit "1" to the digit "2".

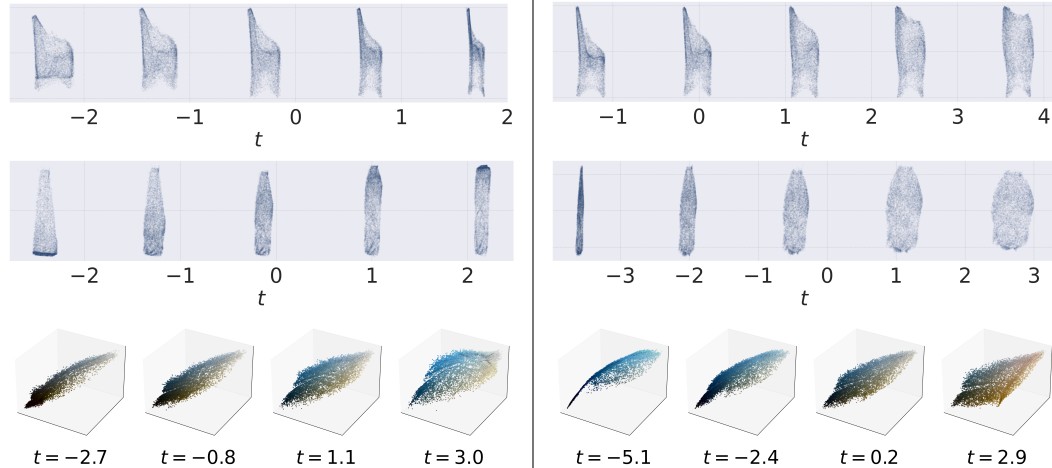

Figure 6: Empirical distributions uniformly sampled along the geodesics corresponding to the first (**left**) and second (**right**) principal components, as computed by GPCAGEN in the 3D point cloud of chairs experiment (**top row**), the 3D point cloud of lamps experiment (**middle row**) and the Landscape images experiment (**bottom row**).

experiment in AppendixA.3 demonstrates how GPCA can be used for outlier detection. For these experiments, $f_\psi$ and $\varphi_\theta$ are MLPs with four hidden layers of size 128 and an output layer of size 1 and $d$ respectively. We found that setting the regularization coefficients $\lambda_I$ and $\lambda_O$ to 1.0 ensures the algorithm works as expected in all experiments. A discussion of the regularization coefficients, along with details on the architecture and hyperparameters, is provided in Appendix E.

**MNIST geodesics.** We represent each image from the MNIST dataset (LeCun et al. (2010)) as a probability measure over $\mathbb{R}^4$. The grayscale pixel intensities define a normalized density over spatial coordinates $(x, y) \in \mathbb{R}^2$, and we further assign each pixel two additional values corresponding to red and blue color channels. We construct two orthogonal geodesics: the first one interpolates between a digit "1" and a digit "2", both assigned a fixed purple by setting the color channels to 0.5. The second one is defined from the midpoint of the first, by linearly interpolating the color from red to blue. As shown in Figures 5 and 9, GPCAGEN successfully recovers the two geodesics intersecting orthogonally. A second experiment on the MNIST dataset is displayed in Appendix A.

**3D point cloud.** We use the ModelNet40 3D point cloud dataset (Wu et al. (2015)) and apply GPCA to a subset of 100 randomly selected lamp point clouds. Figure 6 (**middle row**) and Figure 7 (**left**) demonstrate that the first principal component captures the distinction between hanging lamps (chandeliers) and standing lamps (floor lamps), while the second component reflects variations in the thickness of the lamp structure. We conduct a similar experiment on 100 point clouds from ModelNet40 representing different chairs. As shown in Figure 6 (**top row**) and Figure 10, the first principal component distinguishes between chairs and armchairs, while the second component captures the height of the seat.

**Landscape images.** We took 39 images from the Landscape Pictures dataset (Rougetet (2020)) and use GPCAGEN on the corresponding point clouds, where each point cloud represents color distribution in the image. Figure 6 (**bottom row**) and Figure 7 (**right**) show that the first component captures variations in overall brightness, ranging from dark to bright images, while the second component separates mostly blue images from mostly green ones.

**Baselines** An obvious baseline for GPCAGEN is TPCA. Unlike GPCAGEN, which learns continuous geodesics from empirical distributions of absolutely continuous measures, TPCA acts on discrete measures. A direct numerical comparison between the two methods is therefore not meaningful. However, we include in Appendix A.2 the two principal components returned by TPCA on the 3D point cloud experiments. We observe in Figure 16 that the discrete nature of TPCA produces artifacts, including holes in certain regions and excessive mass concentration in others.

Another natural baseline consists in embedding point clouds into a latent space of dimension $d$ then performing standard PCA on the resulting latent vectors. This approach, in addition to being computationally expensive, does not produce meaningful modes of variation, as shown in Section A.2 of the appendices.

**Code availability.** The code for the Gaussian experiments is available at `https://github.com/alebrigant/bures-wasserstein-gpca`, and the code for the a.c. probability measures is available at `https://github.com/nvesseron`.

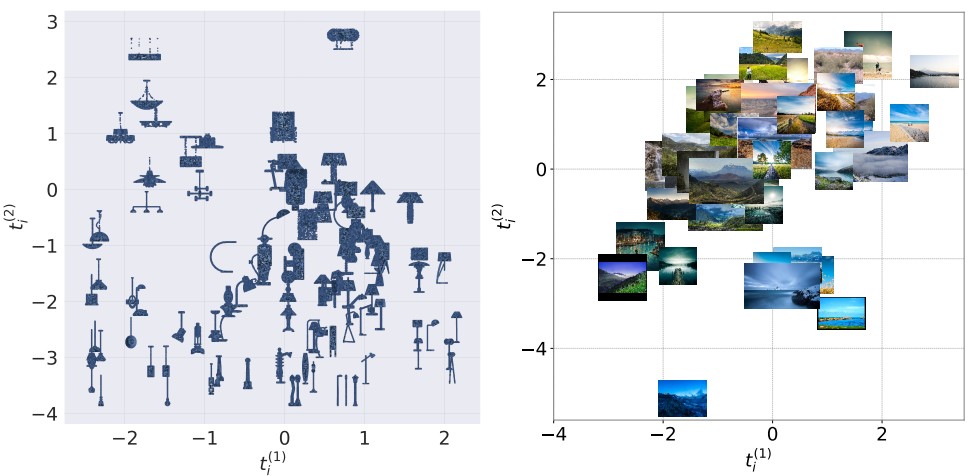

Figure 7: Each lamp point cloud (**left**) and each image (**right**) is embedded in the plane according to its projection times onto the first and second principal components computed by GPCAGEN.

## 6 DISCUSSION

GPCA is a statistical approach for learning the main modes of variations of a set of probability distributions. The first components capture meaningful structure for data lying on a curved space, which then enables downstream tasks such as classification, clustering, and outlier detection. In this work, we have proposed two methods for computing exact GPCA : one tailored for Gaussian distributions and the other for the more general case of a.c. probability distributions. In the Gaussian case, our experiments suggest that GPCA and TPCA generically yield very similar results, except for distributions with covariance matrices that are close to the boundary of the SPD cone, for which GPCA can yield undesirable effects as suggested by the pathological example of Figure 4. In the general case of a.c. probability measures, a key advantage of our approach is that it operates directly on continuous distributions, avoiding the need for empirical approximations of the $\nu_i$, which would require equal sample sizes and can introduce discretization artifacts in the recovered components. Additionally, our method enables sampling from any point along the geodesic components—something not possible with discrete approximations commonly used in TPCA. Otto's parametrization also allowed us to avoid relying on input convex neural networks (ICNNs) by not requiring convex functions, with the trade-off being the need to estimate the eigenvalues of the Hessian of $f$. This perspective opens new directions for parametrizing convex functions without imposing hard architectural constraints.

## ACKNOWLEDGEMENTS

This work benefited from the support of the Agence nationale de la recherche, through the PEPR PDE-AI project (ANR-23-PEIA-0004). This work was performed using HPC resources from GENCI–IDRIS (Grant 2023-103245). This work was partially supported by Hi! Paris through the PhD funding of Nina Vesseron.

## Reproducibility Statement

All implementation details of our proposed method, including model architectures, training procedures, and hyperparameter settings, are provided in Section 5 of the main paper and in Appendix E and D.2. Original theoretical results are presented with complete proofs in Appendix D. The datasets used in our experiments are publicly available.

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

## A    ADDITIONAL EXPERIMENTS AND FIGURES

### A.1    GEODESIC PCA

Here we present additional figures to further explain the experiments described in the paper. Figure 8 concerns the experiment on Gaussian distributions with diagonal covariances described in Section 5.1 corresponding to Figure 4. It shows all three principal components found by tangent PCA (**left**) and geodesic PCA, in two equally optimal solutions (**middle, right**).

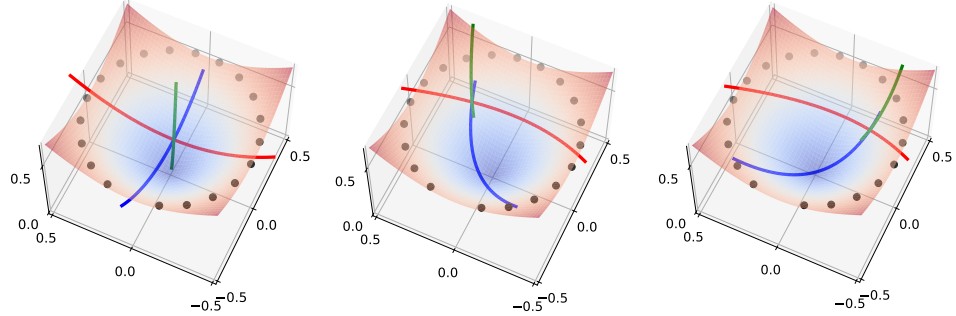

Figure 8: Principal geodesic components of a set of Gaussian distributions whose covariance matrices have same eigenvalues and different orientations, as described in Section 5.1. Tangent PCA yields a unique solution (**left**) where geodesic components cross at the barycenter, while geodesic PCA yields two equally optimal solutions (**middle, right**) where the geodesic components cross at another point. The first geodesic component is shown in red, the second in blue, the third in green.

Figure 9 displays on the plane the two first geodesic components of the MNIST experiment of Section 5.2, while Figure 10 shows the planar representation of the 3D point cloud of chairs experiment given by the projection onto the first two geodesic components found by GPCAGEN algorithm and depicted in Figure 6 (**top row**).

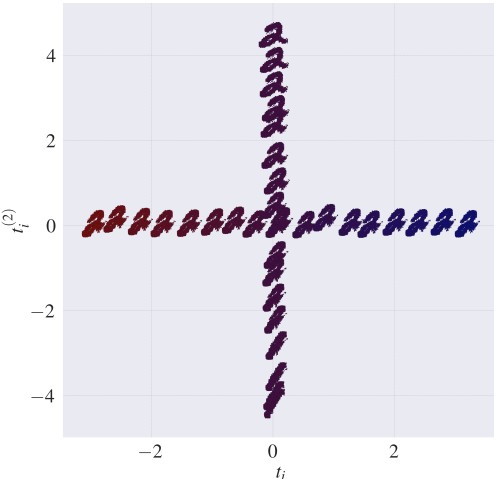

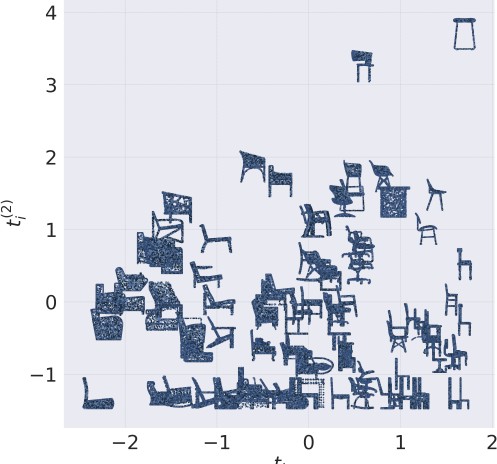

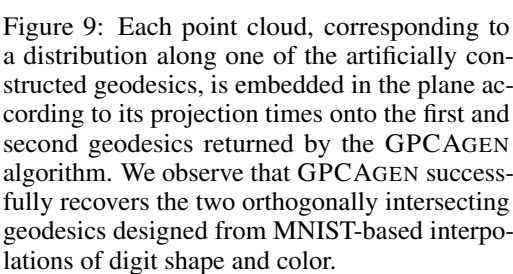

Figure 9: Each point cloud, corresponding to a distribution along one of the artificially constructed geodesics, is embedded in the plane according to its projection times onto the first and second geodesics returned by the GPCAGEN algorithm. We observe that GPCAGEN successfully recovers the two orthogonally intersecting geodesics designed from MNIST-based interpolations of digit shape and color.

Figure 10: Each chair point cloud is embedded in the plane according to its projection times onto the first and second geodesics returned by the GPCAGEN algorithm.

Finally, we present an additional experiment on the MNIST dataset. We use the same color construction as in the experiment presented in Section 5.2, we then apply GPCAGEN to a dataset of 20 red digits "1", 20 blue digits "1", 20 red digits "2", and 20 blue digits "2" (see Figure 12). As shown in Figures 11 and 12, GPCAGEN again identifies two orthogonal geodesics: the first primarily captures variation in color, while the second captures variation in shape—from digit "2" to digit "1".

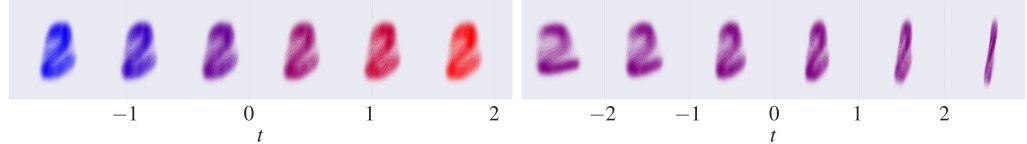

Figure 11: Densities of probability distributions uniformly sampled along the geodesics corresponding to the first and second principal components. The first component (**left**) returned by GPCAGEN captures variation in color space, while the second component (**right**) recovers the interpolation between digit "2" and digit "1".

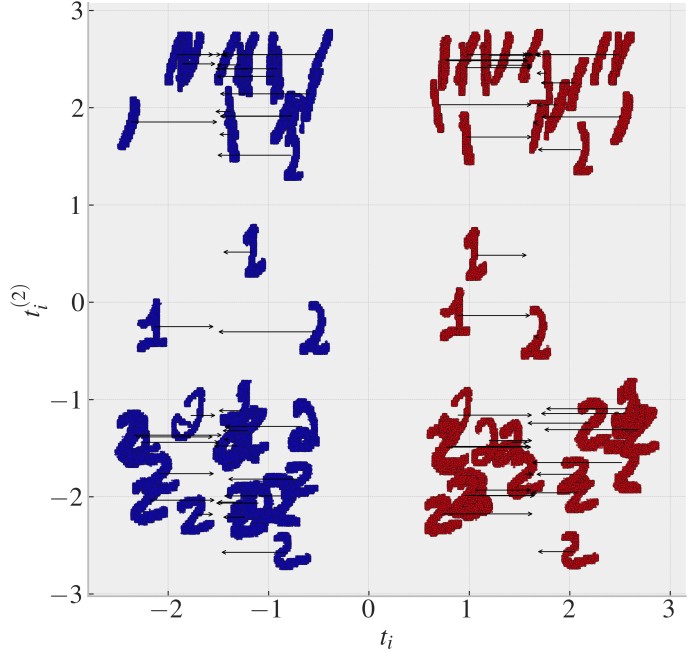

Figure 12: Each MNIST digit is embedded in the plane (the arrows indicate the exact position of each digit) according to its projection times onto the first and second geodesics returned by the GPCAGEN algorithm. We observe that the first principal component recovered by GPCAGEN captures variation in color, while the second component reflects the transformation from digit "2" to digit "1".

## A.2   COMPARISON OF GPCA TO RELATED METHODS

**Other notions of PCA on Gaussian distributions**   There exist a wide variety of metrics on the space of symmetric positive definite matrices, such as e.g. the log-Euclidean, Euclidean-Cholesky or affine-invariant metrics (see Thanwerdas (2022) for a comprehensive overview). Each of these metrics could be used to perform PCA on centered Gaussian distributions. However, there is no obvious quantitative way to compare the results. Each method optimizes its own criterion, and any metric that one could think of to compare the methods would rely on a choice of underlying metric on the space of SPD matrices. Comparison of PCA methods with two different metrics thus boils down to comparing the metrics themselves. We illustrate in Figure 13 the behavior of covariances matrices along geodesics for different metrics.

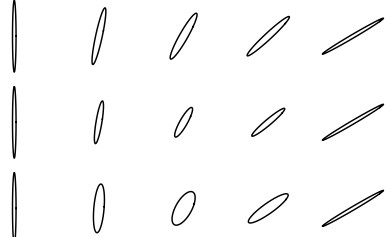

Figure 13: Geodesics on the space of symmetric positive definite matrices from left to right, for (**top**) the Bures-Wasserstein metric, (**middle**) the log-Euclidean metric and (**bottom**) the Euclidean metric on the Cholesky coefficients.

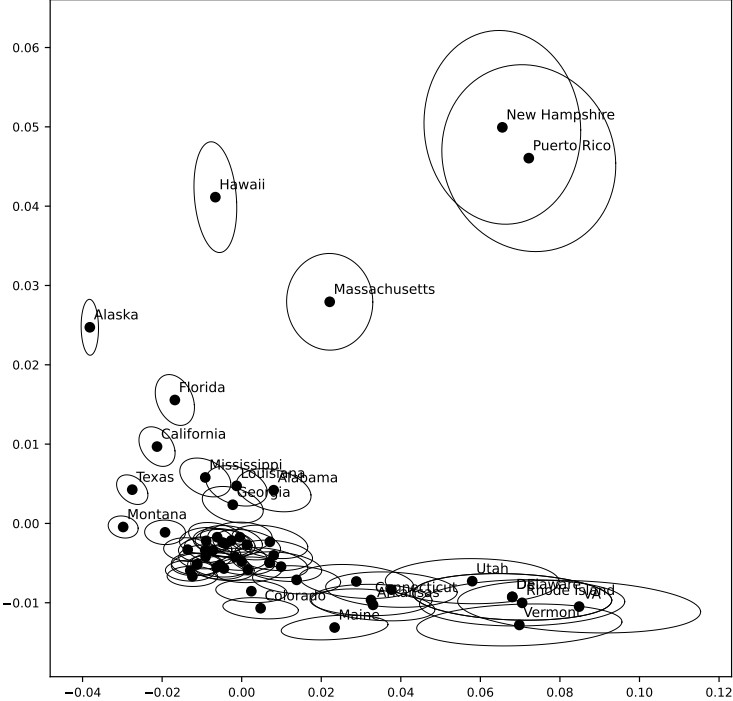

Figure 14: Each dot represents a state projected onto the first two GPCA components computed from the empirical covariance matrices, which are also shown in the figure.

**TPCA on 3D Point Cloud Data**    Here we present the results returned by TPCA on the 3D point cloud experiments, see Figures 16 and 15, and compare them to from those obtained by GPCAGEN.

For the lamps dataset, the first component is similar and captures the distinction between hanging and standing lamps. The second component focuses on the object thickness, like the second GPCAGEN component, but also on whether mass is concentrated at the extremities or the middle of the lamp structure.

For the chairs dataset, both geodesics obtained by TPCA resemble those returned by GPCA. However, the second TPCA component also appears to account for whether the mass is concentrated or not.

Finally, due to the discrete nature of the TPCA algorithm, we observe discretization artifacts in the TPCA components: holes in some parts of the space, mass concentration in others.

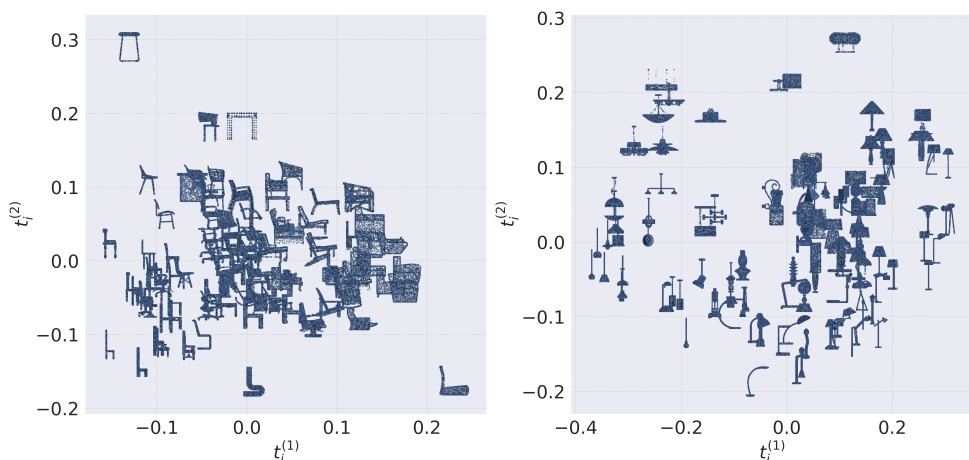

Figure 15: For the chair and the lamp experiment, each point cloud is embedded in the plane according to its projection times onto the first and second principal components computed by TPCA.

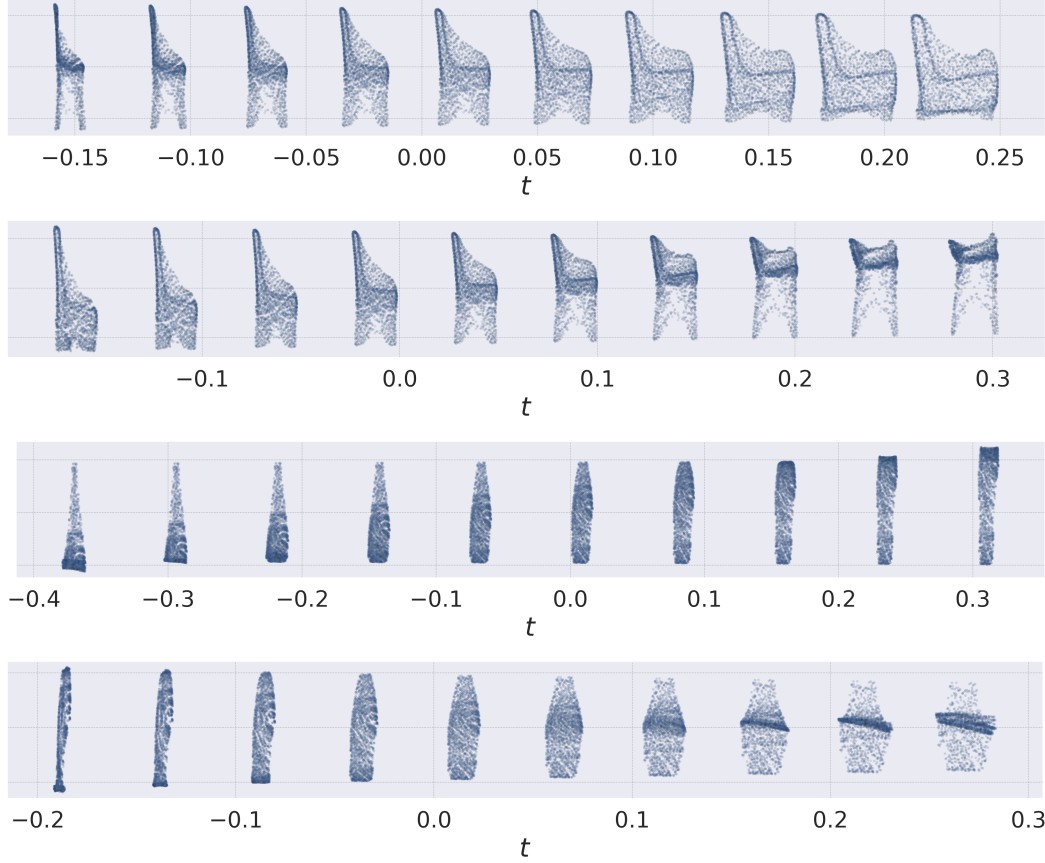

Figure 16: Empirical distributions uniformly sampled along the geodesics corresponding to the first (**first line**) and second (**second line**) principal components, as computed by TPCA in the 3D point cloud of chairs experiment (**top rows**) and the 3D point cloud of lamps experiment (**bottom rows**).

**PCA computed in the latent space of PointNet.** For the 3D point-cloud datasets, we evaluated the natural baseline that consists in embedding point clouds into a latent space of dimension $d$

and then performing standard PCA on the resulting latent vectors. We used a pretrained PointNet autoencoder (Qi et al., 2017) from the public repository `https://github.com/vinits5/pc_autoencoder`, trained on ModelNet40, to encode each point cloud (chairs and lamps) into a $d$-dimensional latent representation, on which PCA was applied. Figure 17 shows the resulting 2D projections. We observe some clustering of similar objects; for example, large lamps tend to group together in the lamp dataset, and chairs versus armchairs form distinguishable clusters. The second principal component for chairs appears to correlate with the height of the seat. Beyond these observations, however, PCA provides limited separability (especially for lamps), and the recovered components are difficult to interpret.

More generally, this approach presents several important limitations:

- Training a point-cloud autoencoder requires a large collection of distributions. In our case (100 distributions), we need to rely on a pretrained autoencoder trained on related dataset.
- PCA on autoencoder embeddings relies heavily on the geometry learned by the encoder. The learned geometry is not guaranteed to align with the Wasserstein structure and the recovered principal components may not reflect meaningful modes of variation (as observed in the experiments above). Moreover, for a given autoencoder that we wish to train, different random seeds at initialization can lead to different learned geometries and thus different PCA components, which is not suitable.

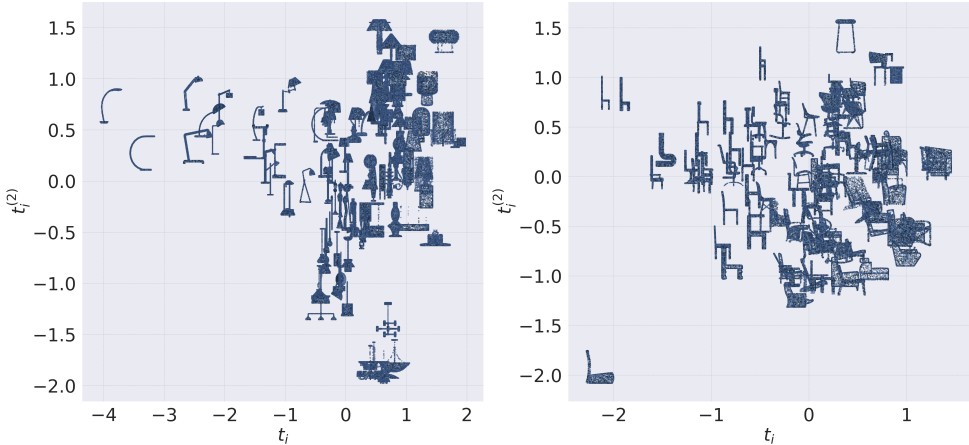

Figure 17: For the lamp (**left**) and the chair (**right**) experiment, each point cloud is embedded in the plane according to its projection times onto the first and second principal components computed by the POINTNET + PCA method.

### A.3 APPLICATION OF GPCA TO OUTLIER DETECTION

In this section, we demonstrate how GPCA can be used for outlier detection. The underlying intuition is that GPCA components capture the structure of the dataset on which they are trained, and samples from a different dataset are expected to lie far from the learned components in Wasserstein distance. In this experiment, we use the ModelNet40 3D point cloud dataset (Wu et al., 2015) and apply GPCA to a subset of 100 randomly selected chair point clouds to compute the first two components. For a new point cloud $X$, we define its score as the sum of the Wasserstein distances between $X$ and its projections onto the first two learned GPCA components. To compute the Wasserstein distance between $X$ and a component, we use `ot.emd` from the POT library. Specifically, for each component, we perform a grid search over 20 equally spaced values of $t$ between $t_{\min}$ and $t_{\max}$, computing the Wasserstein distance between $X$ and 2048 samples drawn from the component at each $t$, and select the $t$ that minimizes this distance. We repeat the same procedure for the second component and sum the two minimal distances to obtain the final score.

We evaluate this approach on 120 point clouds: 60 new chairs (not used for training) and 60 point clouds of cars. The left histogram in Figure 18 shows the resulting scores. We observe that the scores

of the chair point clouds (in blue) are lower than those of the car point clouds (in green), indicating that it is possible to detect whether a point cloud is not a chair using this score. We also repeat the experiment with 60 point clouds of planes, shown in the right histogram of Figure 18, and observe that the separation between chair and plane scores is even more pronounced.

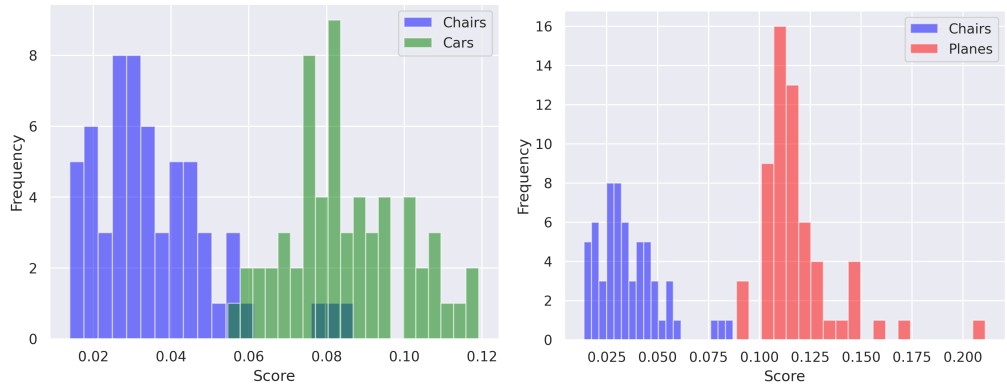

Figure 18: GPCA scores obtained on 60 **new** point clouds of chairs (never seen during training) and 60 point clouds of cars (**left**) / planes (**right**). The separation of the histograms indicates that GPCA can be used for outlier detection.

## B    THE OTTO-WASSERSTEIN GEOMETRY

In this section, we briefly describe the fiber bundle structure over the Wasserstein space due to Otto (2001), that is behind the Riemannian interpretation of the Wasserstein distance. We then present its restriction to the space of centered non-degenerate Gaussian distributions, which coincides with the Bures-Wasserstein Riemannian geometry on SPD matrices. Finally, we relate Otto's parametrization of geodesics to McCann's interpolation.

We present these well-known results without proofs and refer the interested reader to Otto (2001); Khesin et al. (2021) and (Ambrosio et al., 2013, Section 6.1) for more details in the general setting and to Takatsu (2011); Malagò et al. (2018); Bhatia et al. (2019) for details and proofs in the Gaussian setting.

### B.1    THE OTTO-WASSERSTEIN GEOMETRY OF A.C. DISTRIBUTIONS

Consider the space $\mathrm{Prob}(\Omega)$ of absolutely continuous probability measures with smooth densities with respect to the Lebesgue measure, and support included in a compact set $\Omega \subset \mathbb{R}^d$, as well as the space $\mathrm{Diff}(\Omega)$ of diffeomorphisms on $\Omega$. These spaces can be equipped with an infinite-dimensional manifold structure, see e.g. Ebin & Marsden (1970), that we will not describe here. The tangent space of $\mathrm{Diff}(\Omega)$ at $\varphi \in \mathrm{Diff}(\Omega)$ is given by

$$T_\varphi \mathrm{Diff}(\Omega) = \{v \circ \varphi, \ v : \Omega \to \mathbb{R}^d \text{ vector field}\}.$$

We fix a reference measure $\rho \in \mathrm{Prob}(\Omega)$ and equip $\mathrm{Diff}(\Omega)$ with the $L^2$-metric with respect to $\rho$, defined for any tangent vectors $u \circ \varphi, v \circ \varphi \in T_\varphi \mathrm{Diff}(\Omega)$ as

$$\langle u \circ \varphi, v \circ \varphi \rangle_{L^2(\rho)} : = \int (u \circ \varphi) \cdot (v \circ \varphi) \, d\rho = \int u \cdot v \, d\mu,$$

where $\mu = \varphi_\# \rho$. Then the space of diffeomorphisms can be decomposed into *fibers*, defined to be equivalence classes under the projection

$$\pi \colon \mathrm{Diff}(\Omega) \to \mathrm{Prob}(\Omega), \quad \varphi \mapsto \varphi_\# \rho.$$

Specifically, the fiber over $\mu \in \mathrm{Prob}(\Omega)$ is given by $\pi^{-1}(\mu) = \{\varphi \in \mathrm{Diff}(\Omega), \varphi_\# \rho = \mu\}$, see Figure 19 (**right**). The tangent space to the fiber $\pi^{-1}(\mu)$ at $\varphi \in \mathrm{Diff}(\Omega)$ and its orthogonal with respect to the $L^2(\rho)$-metric are refered to as the *vertical* and *horizontal* spaces respectively :

$$\mathrm{Ver}_\varphi : = \ker d\pi_\varphi, \quad \mathrm{Hor}_\varphi := (\mathrm{Ver}_\varphi)^\perp,$$

where $d\pi_\varphi\colon T_\varphi \mathrm{Diff}(\Omega) \to T_{\pi(\varphi)}\mathrm{Prob}(\Omega)$ denotes the differential of $\pi$ at $\varphi$. Moving along vertical vectors in $\mathrm{Diff}(\Omega)$ means staying in the same fiber, i.e. projecting always to the same measure $\mu$ in the bottom space. On the contrary, moving along horizontal vectors means moving orthogonally to the fibers, i.e., in the direction that gets fastest away from the fiber. The following proposition gives the form of vertical and horizontal vectors.

**Proposition 6.** *Let $\varphi \in \mathrm{Diff}(\Omega)$. Then*

$$\mathrm{Ver}_\varphi = \{w \circ \varphi,\ \nabla \cdot (w\mu) = 0\},$$
$$\mathrm{Hor}_\varphi = \{\nabla f \circ \varphi,\ f \in C^\infty(\Omega)\}.$$

The following results state that line segments and $L^2(\rho)$-distances in $\mathrm{Diff}(\Omega)$ can be used to compute Wasserstein geodesics and distances in the space of probability measures $\mathrm{Prob}(\Omega)$, provided we restrict to horizontal displacements.

**Proposition 7.** *The projection $\pi\colon \mathrm{Diff}(\Omega) \to \mathrm{Prob}(\Omega)$ is a Riemannian submersion, i.e. $d\pi_\varphi\colon \mathrm{Hor}_\varphi \to T_{\pi(\varphi)}\mathrm{Prob}(\Omega)$ is an isometry for any $\varphi \in \mathrm{Diff}(\Omega)$.*

This implies the following.

**Proposition 8** (Proposition 2 in main). *Any geodesic $t \mapsto \mu(t)$ for the Wasserstein metric in equation 2 is the $\pi$-projection of a line segment in $\mathrm{Diff}(\Omega)$ going through a diffeomorphism $\varphi$ at horizontal speed $\nabla f \circ \varphi$ for some smooth function $f \in \mathcal{C}(\mathbb{R}^d)$. That is, for $t$ defined in a certain interval $(t_{min}, t_{max})$,*

$$\mu(t) = \pi(\varphi + t\nabla f \circ \varphi) = (\mathrm{id} + t\nabla f)_\#(\varphi_\# \rho). \tag{17}$$

*Another geodesic $\tilde{\mu}(t) = \pi(\varphi + t\nabla \tilde{f} \circ \varphi)$ is orthogonal to $\mu(t)$ at $t = 0$ for the Riemannian metric inducing the Wasserstein distance if and only if $\langle \nabla f \circ \varphi, \nabla \tilde{f} \circ \varphi \rangle_{L^2(\rho)} = 0$.*

We comment on the link between this parametrization and McCann's interpolation in Section B.3.

## B.2 THE OTTO-WASSERSTEIN GEOMETRY OF GAUSSIAN DISTRIBUTIONS

The Bures-Wasserstein distance in equation 3 on the space $S_d^{++}$ of symmetric positive definite (SPD) matrices is the geodesic distance induced by a Riemannian metric $g^{BW}$, which can be written in different ways. Here we use the expression from (Thanwerdas, 2022, Table 4.7), defined for $\Sigma = PDP^\top \in S_d^{++}$ and $U = PU'P^\top \in S_d$, by

$$g_\Sigma^{BW}(U,U) = \frac{1}{2} \sum_{1 \le i,j \le d} \frac{1}{d_i + d_j} {U'_{ij}}^2, \tag{18}$$

where the $d_i$'s are the diagonal elements of $D$. The associated Riemannian geometry can be described by Otto's fiber bundle restricted to the space of centered Gaussian distributions, in the following way.

In this setting, diffeomorphisms are restricted to invertible linear maps $\varphi\colon u \mapsto Au$ for some invertible matrix $A$, i.e. the space of diffeomorphisms is replaced by the Lie group of invertible matrices $GL_d$. Tangent vectors are then given by linear maps $u \mapsto Xu$ for any matrix $X \in \mathbb{R}^{d \times d}$. Fixing the standard normal distribution $\rho = \mathcal{N}(0, \mathrm{Id})$ as reference measure, the $L^2$-metric with respect to $\rho$ between $u \mapsto Xu$ and $u \mapsto Yu$ is then written, for any $X, Y \in \mathbb{R}^{d \times d}$:

$$\int_{\mathbb{R}^d} \varphi(u)^\top \psi(u) d\rho(u) = \int_{\mathbb{R}^d} \mathrm{tr}(\varphi(u)\psi(u)^\top) d\rho(u) = \mathrm{tr}\left( \int_{\mathbb{R}^d} Xuu^\top Y^\top d\rho(u) \right) = \mathrm{tr}(XY^\top),$$

yielding the standard Frobenius inner product on (the tangent space of) $GL_d$. We obtain a fibration of the top space $GL_d$ over the bottom space $S_d^{++}$ by considering the following projection

$$\pi\colon GL_d \to S_d^{++}, \quad A \mapsto AA^\top, \tag{19}$$

see Figure 19 (**left**). The fiber over $\Sigma \in S_d^{++}$ is

$$\pi^{-1}(\Sigma) = \{A \in GL_d,\ AA^\top = \Sigma\} = \Sigma^{1/2}O_d, \tag{20}$$

where $O_d$ denotes the space of orthogonal matrices and $\Sigma^{1/2}$ denotes the only SPD square root of the SPD matrix $\Sigma$. The differential of the projection $\pi(A) = AA^\top$ is given by

$$d\pi_A(X) = XA^\top + AX^\top. \tag{21}$$

Therefore, vertical vectors, which are those tangent to the fibers, or equivalently, those belonging to the kernel of $d\pi_A(X)$, are given by

$$
\begin{aligned}
\mathrm{Ver}_A\colon\ &= \{X \in \mathbb{R}^{d\times d},\ XA^\top + AX^\top = 0\} \\
&= \{X \in \mathbb{R}^{d\times d},\ XA^\top \text{ is antisymmetric}\} \\
&= \{X = K(A^\top)^{-1},\ K \in S_d^\perp\} = S_d^\perp (A^\top)^{-1}.
\end{aligned}
$$

where $S_d^\perp$ denotes the space of antisymmetric matrices of size $d$. Once again, moving along vertical vectors in $GL_d$ means staying in the same fiber, i.e. projecting always to the same SPD matrix in the bottom space $S_d^{++}$. Horizontal vectors are those that are orthogonal to all vertical vectors (for the Frobenius metric), i.e. matrices $X$ such that for any antisymmetric matrix $K$:

$$
0 = \langle X, K(A^\top)^{-1}\rangle = \mathrm{tr}(XA^{-1}K^\top)
$$

which is equivalent to $XA^{-1}$ symmetric (this can be seen by taking for $K$ the basis elements of $S_d^\perp$ in the above equation), yielding

$$
\begin{aligned}
\mathrm{Hor}_A\colon\ &= \{X \in \mathbb{R}^{d\times d},\ (A^\top)^{-1}X^\top = XA^{-1}\} \\
&= \{X \in \mathbb{R}^{d\times d},\ X^\top A - A^\top X = 0\} \\
&= \{X = KA,\ K \in S_d\} = S_d A
\end{aligned}
$$

where $S_d$ denotes the space of symmetric matrices.

**Proposition 9.** *The projection $\pi\colon GL_d \to S_d^{++}$, $A \mapsto AA^\top$ is a Riemannian submersion, i.e. $d\pi_A$ is an isometry from $\mathrm{Hor}_A$ equipped with the Frobenius inner product to $T_{\pi(A)}S_d^{++}$ equipped with the inner product $g_{\pi(A)}^{BW}$, for any $A \in GL_d$.*

Just like in the general case, this yields a way to lift the computation of geodesics and distances.

**Proposition 10** (Propositon 1 in main)**.** *Any geodesic $t \mapsto \Sigma(t)$ in $S_d^{++}$ for the Bures-Wasserstein metric in equation 3 is the $\pi$-projection of a horizontal line segment in $GL_d$, that is*

$$
\Sigma(t) = \pi(A + tX) = (A + tX)(A + tX)^\top, \quad A \in GL_d,\ X \in \mathrm{Hor}_A, \tag{22}
$$

*where $t$ is defined in a certain time interval $(t_{min}, t_{max})$. Also, the Bures-Wasserstein distance between two covariance matrices $\Sigma_1, \Sigma_2 \in S_d^{++}$ is given by the minimal distance between their fibers*

$$
BW_2(\Sigma_1, \Sigma_2) = \inf_{Q_1,Q_2 \in O_d} \|\Sigma_1^{1/2}Q_1 - \Sigma_2^{1/2}Q_2\| = \inf_{Q \in SO_d} \|\Sigma_1^{1/2} - \Sigma_2^{1/2}Q\|, \tag{23}
$$

*where $\|\cdot\|$ is the Frobenius norm and $SO_d$ is the special orthogonal group.*

Formula in equation 22 and the first equality of equation 23 are direct consequences of the fact that $\pi$ is a Riemannian submersion. To obtain the second equality of equation 23, we first notice that optimizing on $Q_1, Q_2 \in O_d$ is equivalent to optimizing on a single $Q \in O_d$ thanks to the invariance of the Frobenius metric w.r.t. the right action of $O_d$. And second, that the infimum is attained at (see (Bhatia et al., 2019, Equations 3 and 35))

$$
Q^* = \Sigma_2^{-1/2}T\Sigma_1^{1/2}, \quad \text{where} \quad T = \Sigma_1^{-1/2}(\Sigma_1^{1/2}\Sigma_2\Sigma_1^{1/2})^{1/2}\Sigma_1^{-1/2}
$$

is the Monge map from $\Sigma_1$ to $\Sigma_2$ (see (Malagò et al., 2018, equation 8)), and so $Q^*$ has positive determinant and belongs to $SO_d$.

Thus the closest element of the fiber $\pi^{-1}(\Sigma_2)$ to $\Sigma_1^{1/2}$ is given by $\Sigma_2^{1/2}Q^* = T\Sigma_1^{1/2}$, i.e. by left multiplying $\Sigma_1^{1/2}$ by the Monge map $T$. This is more generally true for any representative of $\Sigma_1$:

**Proposition 11.** *Let $\Sigma_1, \Sigma_2 \in S_d^{++}$, $T$ the Monge map from $\Sigma_1$ to $\Sigma_2$, $A_1 \in \pi^{-1}(\Sigma_1)$. Then $A_2 := TA_1$ is said to be* aligned *with respect to $A_1$, that is, it is the closest point in $\pi^{-1}(\Sigma_2)$ to $A_1$. More precisely, we have*

1.  $A_2 - A_1 = (T - I)A_1 \in \mathrm{Hor}_{A_1}$

2.  $\mathrm{Log}_{\Sigma_1}(\Sigma_2)\colon = d\pi_{A_1}((T - I)A_1) = (T - I)\Sigma_1 + \Sigma_1(T - I)$

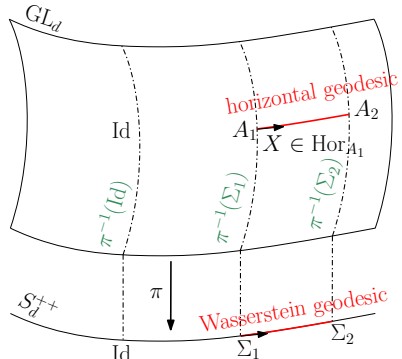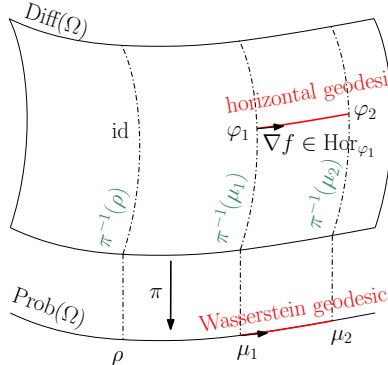

Figure 19: The Otto-Wasserstein geometry of (**left**) centered Gaussian distributions and (**right**) a.c. probability distributions. Figures inspired by Khesin et al. (2021).

3. $BW_2(\Sigma_1, \Sigma_2) = \|\text{Log}_{\Sigma_1} \Sigma_2\|^{BW}_{\Sigma_1} = \|(T - I)A_1\|$

*where* $\text{Log}$ *is the Riemannian logarithm map,* $\| \cdot \|^{BW}_\Sigma = \sqrt{g^{BW}_\Sigma(\cdot, \cdot)}$ *and* $\| \cdot \|$ *is the Frobenius norm.*

This means that to compute the Bures-Wasserstein distance between two covariance matrices $\Sigma_1$ and $\Sigma_2$, one can consider any representative $A_1$ in the fiber over $\Sigma_1$, compute the representative $A_2$ of $\Sigma_2$ aligned to $A_1$ (using the Monge map) and finally compute the Frobenius norm of $A_2 - A_1$.

### B.3 GEODESIC PARAMETRIZATION

There are two classical parameterizations for Wasserstein geodesics in the space of a.c. probability measures.

**McCann's interpolation** The first one, due to McCann (1997), is given between two probability distributions $\mu_0$ and $\mu_1$, and depends on the optimal transport map in equation 2, obtained as the gradient of a convex function $u$, that is $T^{\mu_1}_{\mu_0} = \nabla u$ and

$$\mu_t = ((1 - t)\,\text{id} + t\nabla u)_\# \mu_0 = (\text{id} + t(\nabla u - \text{id}))_\# \mu_0, \quad t \in [0, 1]. \tag{24}$$

**Otto's geodesic** The second one, exploiting Otto's fiber bundle geometry in Otto (2001), consists in writing a geodesic in the Wasserstein space as the projection of a horizontal geodesic in the total space of diffeomorphisms. Such a horizontal geodesic is a line segment going through a diffeomorphism $\varphi$ with a horizontal speed $\nabla f \circ \varphi$, where $f$ is any smooth function (not necessarily convex). Therefore we get

$$\mu_s = (\varphi + s\nabla f \circ \varphi)_\# \rho = (\text{id} + s\nabla f)_\#(\varphi_\# \rho), \quad s \in (s_0, s_1). \tag{25}$$

In this second expression, the bounds on the time $s$ depends on the function $f$. Indeed, for $\mu_s$ to be a geodesic, $\text{id} + s\nabla f$ needs to remain is the space of diffeomorphisms for a given $s$, which means that $\text{id} + s\text{Hess}\,f$ needs to be positive definite. Therefore, we get the following conditions depending on the minimum $\lambda_{\min}$ and maximum $\lambda_{\max}$ eigenvalues of $\text{Hess}\,f$:

$$\begin{cases} s \in (-\infty, -1/\lambda_{\min}) & \text{if} \quad \lambda_{\max} < 0, \\ s \in (-1/\lambda_{\max}, +\infty) & \text{if} \quad \lambda_{\min} > 0, \\ s \in (-1/\lambda_{\max}, -1/\lambda_{\min}) & \text{if} \quad \lambda_{\min} < 0 < \lambda_{\max}. \end{cases} \tag{26}$$

It is clear that equation 24 is a particular case of equation 25, where we choose $\varphi_\# \rho = \mu_0$ and $\nabla f = \nabla u - \text{id}$. Conversely, one can write equation 25 under the form of equation 24. For a given diffeomorphism $\varphi$ and function $f$, consider the geodesic given by equation 25, and set $\mu_0 = \varphi_\# \rho$. Assume that we are in the case where all eigenvalues of $\text{Hess}\,f$ are negative, then $s$ must be in $]-\infty, -1/\lambda_{\min}[$. Consider $s^* \in\,]0, -1/\lambda_{\min}[$, and define $\mu_1 := \mu_{s^*} = (\text{id} + s^*\nabla f)_\# \mu_0$. Setting $t = s/s^*$ we have that the geodesic between $\mu_0$ and $\mu_1$ is written

$$\mu_t = (\text{id} + ts^*\nabla f)_\# \mu_0 = (\text{id} + t(\nabla u - \text{id}))_\# \mu_0, \quad t \in [0, 1].$$

for $u(x) = s^*f + \|x\|^2/2$. Now for any eigenvalue $\lambda_i$ of $H_f$ the Hessian of $f$, we have

$$\lambda_i > \lambda_{\min} > -1/s^* \quad \text{i.e.} \quad s^*\lambda_i + 1 > 0.$$

by the interval of definition of $s^*$. This means that the Hessian $H_u = s^*H_f + \mathrm{id}$ is positive definite, which means that $u$ is necessarily convex. The other cases work similarly.

**The Gaussian case**   Transposing Otto's formulation in equation 25 to the case of a geodesic between Gaussian distributions means that for $A \in GL_d$ and $X \in \mathrm{Hor}_A$ such that $\|X\| = 1$, the interval of definition of a geodesic depends on the invertibility of $A + sX$. In turn, the maximal interval of definition of $s \in (s_0, s_1)$ is defined from the eigenvalues of $XA^{-1}$, through the same formula in equation 26.

## C   LINEARIZED OPTIMAL TRANSPORT AND TANGENT PCA

In this section, we provide the definition of linearized Wasserstein distance and details on how to perform tangent PCA for both Gaussian distributions and general a.c. distributions. Tangent PCA is a widely used approach to compute PCA on the Wasserstein space, that consists in embedding probability distributions into the tangent space at some reference measure $\rho$, and performing PCA in the tangent space with respect to the linearized Wasserstein distance.

### C.1   THE CASE OF CENTERED GAUSSIAN DISTRIBUTIONS

We consider $n$ covariance matrices $\Sigma_1, \ldots, \Sigma_n$ and their Bures-Wasserstein barycenter (or Fréchet mean) $\bar{\Sigma}$, that is, the SPD matrix verifying (see Agueh & Carlier (2011)):

$$\bar{\Sigma} = \underset{\Sigma \in S_d^{++}}{\arg\min} \sum_{i=1}^{n} BW_2^2(\Sigma, \Sigma_i). \tag{27}$$

The idea behind tangent PCA is to represent each data point by the corresponding tangent vector, given by the Riemannian logarithm map, in the tangent space at the reference point $\bar{\Sigma}$, i.e.

$$\{\mathrm{Log}_{\bar{\Sigma}}\Sigma_i\}_{i=1}^{n} \subset T_{\bar{\Sigma}}S_d^{++}. \tag{28}$$

Now, one can lift the computations from the tangent space at $\bar{\Sigma}$ to the horizontal space at a point in the fiber over $\bar{\Sigma}$, say $A := \bar{\Sigma}^{1/2}$, by aligning all representatives to $A$, see Proposition 11. The key point is that the tangent space at $\bar{\Sigma}$ equipped with the Bures-Wasserstein Riemannian metric is isometric to $\mathrm{Hor}_A := S_dA$ equipped with the Frobenius inner product – where we recall that $S_d$ is the space of symmetric matrices. This means that instead of performing PCA for the Bures-Wasserstein inner product on the tangent vectors in equation 28, we can instead perform linear PCA on their pre-images by $d\pi_A$, see Proposition 11:

$$\{(T_i - I)A\}_{i=1}^{n} \subset \mathrm{Hor}_{A_1}, \quad \text{where} \quad T_i = \Sigma_i^{-1/2}(\Sigma_i^{1/2}\bar{\Sigma}\Sigma_i^{1/2})^{1/2}\Sigma_i^{-1/2}.$$

$T_i$ is the optimal transport map from $\bar{\Sigma}$ to $\Sigma_i$, see Section B.2. Now, noticing that

$$\langle K_1A, K_2A \rangle = \mathrm{Tr}(K_1AA^\top K_2^\top) = \mathrm{Tr}(K_1\bar{\Sigma}K_2^\top), \quad \forall K_1, K_2 \in S_d,$$

we see that the space $\mathrm{Hor}_A$ equipped with the Frobenius inner product is itself isometric to $S_d$ equipped with the Frobenius inner product weighted by $\bar{\Sigma}$. Therefore, tangent PCA is performed through Euclidean PCA on the (centered) vectors $\{T_i - I\}_{i=1}^{n}$, in the vector space $S_d$, with respect to the Frobenius metric weighted by $\bar{\Sigma}$. Another way to see this is by noticing that the linearized Bures-Wasserstein distance $BW_{2,\bar{\Sigma}}$ with respect to $\bar{\Sigma}$ is given by

$$\begin{aligned} BW_{2,\bar{\Sigma}}(\Sigma_1, \Sigma_2) &:= \|\mathrm{Log}_{\bar{\Sigma}}\Sigma_1 - \mathrm{Log}_{\bar{\Sigma}}\Sigma_2\|_{\bar{\Sigma}}^{BW} \\ &= \|d\pi_{\bar{\Sigma}^{1/2}}((T_1 - I)\bar{\Sigma}^{1/2}) - d\pi_{\bar{\Sigma}^{1/2}}((T_2 - I)\bar{\Sigma}^{1/2})\|_{\bar{\Sigma}}^{BW} \\ &= \|(T_1 - I)\bar{\Sigma}^{1/2} - (T_2 - I)\bar{\Sigma}^{1/2}\| \\ &= \|(T_1 - T_2)\bar{\Sigma}^{1/2}\| \end{aligned}$$

where $\|\cdot\|^{BW}$ denotes the norm associated to the Bures Wasserstein Riemannian metric in equation 18, $\pi$ is Otto's projection in equation 19, and we have used Propositions 9 and 11. Finally,

$$BW_{2,\bar{\Sigma}}(\Sigma_1, \Sigma_2) := \|\mathrm{Log}_{\bar{\Sigma}}\Sigma_1 - \mathrm{Log}_{\bar{\Sigma}}\Sigma_2\|_{\bar{\Sigma}}^{BW} = \|T_1 - T_2\|_{\bar{\Sigma}}, \tag{29}$$

where $\|\cdot\|_{\bar{\Sigma}}$ denotes the Frobenius norm weighted by $\bar{\Sigma}$.

## C.2 THE CASE OF A.C. DISTRIBUTIONS

Similarly, one can embed a.c. probability distributions $\nu_1, \ldots, \nu_n$ into the $L^2(\rho)$ space at some a.c. reference measure $\rho$ through the optimal maps $\nu_i \mapsto T_\rho^{\nu_i}$ in the Monge problem in equation 2. Then, the Wasserstein distance can be approximated by the linearized Wasserstein distance in Wang et al. (2013) given by

$$W_{2,\rho}(\nu_1, \nu_2) = \|T_\rho^{\nu_1} - T_\rho^{\nu_2}\|_{L^2(\rho)}. \tag{30}$$

Note that as previously mentioned, this metric induces distortions : while the radial distances from $\rho$ to any $\mu_i$ are preserved, that is $\|\mathrm{id} - T_\rho^{\nu_i}\|_{L^2(\rho)} = W_2(\rho, \nu_i)$, other distances are not $\|T_\rho^{\nu_1} - T_\rho^{\nu_2}\|_{L^2(\rho)} \neq W_2(\nu_1, \nu_2)$. A recent paper by Letrouit & Mérigot (2024) proved however, that under some assumptions, $W_{2,\rho}$ is bi-Hölder equivalent to $W_2$, which indicates that the distortion effect can be controlled.

Then, denoting $\bar{\nu}_n$ the Wasserstein barycenter as in Agueh & Carlier (2011) of $\nu_1, \ldots, \nu_n$, that is the solution of

$$\bar{\nu}_n \in \arg\min_\nu \sum_{i=1}^n W_2^2(\nu, \nu_i), \tag{31}$$

tangent PCA consists in performing classical PCA, see e.g. Ramsay & Silverman (2002), of $(T_{\bar{\nu}_n}^{\nu_i} - \mathrm{id})_{i=1}^n$ in the Hilbert space $L^2(\bar{\nu}_n)$.

# D GEODESIC PCA FOR GAUSSIAN DISTRIBUTIONS

In this section, we present the proofs related to geodesic PCA for Gaussian distributions and the implementation of our algorithm in this case.

## D.1 PROOFS RELATED TO GPCA FOR GAUSSIAN DISTRIBUTIONS

We first prove the existence of mimimizers for the GPCA problems lifted to Otto's fiber bundle.

**Lemma 1.** *The GPCA problem in equation 12 for the first component admits a global minimum.*

*Proof.* First, let us define the set of normalized matrices $\mathbb{B} := \{X \in \mathbb{R}^{d \times d}, \|X\| = 1\}$. By denoting $\lambda_{\min}$ (resp. $\lambda_{\max}$) the smallest (resp. largest) eigenvalue of $XA^{-1}$, extending the geodesic $t \mapsto A + tX$ as far as possible (see Section B.3) means that the closed interval $[t_{\min}, t_{\max}]$ is defined for some fixed $\varepsilon > 0$ by

$$\begin{cases} (-\infty, -1/\lambda_{\min} - \varepsilon] & \text{if} \quad \lambda_{\max} < 0, \\ [-1/\lambda_{\max} + \varepsilon, +\infty) & \text{if} \quad \lambda_{\min} > 0, \\ [-1/\lambda_{\max} + \varepsilon, -1/\lambda_{\min} - \varepsilon] & \text{if} \quad \lambda_{\min} < 0 < \lambda_{\max}. \end{cases} \tag{32}$$

Let us now consider the function

$$F \colon GL_d \times \mathbb{B} \times (\mathbb{R}^{d \times d})^n \longrightarrow \mathbb{R}$$

$$(A, X, (Q_i)_{i=1}^n) \longmapsto \sum_{i=1}^n \|A + p_{(A,X)}(t_i)X - \Sigma_i^{1/2}Q_i\|^2 =: \sum_{i=1}^n g_i(A, X, Q_i),$$

where $t_i = \langle \Sigma_i^{1/2}Q_i - A, X \rangle$ and $p_{(A,X)} \colon \mathbb{R} \to \mathbb{R}$ is the projection operator that clips a point $t$ into $[t_{\min}, t_{\max}]$, which depends on $A$ and $X$. Then the function $F$ is continuous on $GL_d \times \mathbb{B} \times (\mathbb{R}^{d \times d})^n$ as composition of linear and continuous functions. Note that the function $(A, X) \mapsto p_{(A,X)}(t_i)$ is continuous by eigenvalue continuity, see Li & Zhang (2019). Additionally, the function $F$ is coercive (see e.g. Zalinescu (2002)) on $GL_d \times \mathbb{B} \times (\mathbb{R}^{d \times d})^n$. Indeed, on a diagonal $\{A = \Sigma_i^{1/2}Q_i, \text{ for } (A, Q_i) \in GL_d \times \mathbb{R}^{d \times d}\}$ for some $i \in \{1, \ldots, n\}$, we have $t_i = 0$, and therefore we have either $g_i(A, X, Q_i) = 0$ if $p_{(A,X)}(0) = 0$, or $g_i(A, X, Q_i) = \varepsilon\|X\|^2 = \varepsilon$ otherwise. This would imply that $g_i(A, X, Q_i)$ doesn't go to infinity when the norm $\|(A, X, Q_i)\| \to \infty$. However, in this case, we have $g_j(A, X, Q_j) \to \infty$ when $\|(A, X, Q_j)\| \to \infty$ for any $j \neq i$. Moreover, as $p_{(A,X)}(t_i)$ is a clipping, it won't play a role in the coercivity. We conclude by the fact that the function $(A, X) \mapsto X^\top A - A^\top X$ is continuous, implying that the set of constraint $\{(A, X) \in GL_d \times \mathbb{R}^{d \times d} : X^\top A - A^\top X = 0\}$ is closed and $\mathbb{B}$ and $SO_d$ are compact. The optimization problem in equation 12 thus admits a global minimum. $\square$

Note that this result also applies for the second component in equation 13 and the higher order components.

**Proposition 12** (Proposition 3 in main). *Let $\pi\colon GL_d \to S_d^{++}$, $A \mapsto AA^\top$ and $(A_1, X_1, (Q_i)_{i=1}^n)$ be a solution of*

$$\inf \ F(A_1, X_1, (Q_i)_{i=1}^n)\colon \ = \sum_{i=1}^n \|A_1 + p_{A_1, X_1}(t_i)X_1 - \Sigma_i^{1/2}Q_i\|^2,$$

*subject to* $\quad A_1 \in GL_d, \ X_1 \in \mathrm{Hor}_{A_1}, \ \|X_1\|^2 = 1, \ Q_1, \ldots, Q_n \in SO_d.$

*Then there exist $t_{min}, t_{max} \in \mathbb{R}$ such that the geodesic $\Sigma\colon t \in [t_{min}, t_{max}] \mapsto \pi(A_1 + tX_1)$ in $S_d^{++}$ minimizes equation 11.*

*Proof.* A horizontal geodesic in $GL_d$ is a straight line going through a base point $A \in GL_d$ in the direction of a horizontal vector $X \in \mathrm{Hor}_A$ (that we consider normalized, ie. $\|X\|^2 = 1$), i.e. $t \mapsto A + tX \in GL_d$. Denoting $[t_{\min}, t_{\max}]$ the interval constructed in equation 32 which depends on the eigenvalues of $XA^{-1}$, we have that $(\pi(A + tX))_{t \in [t_{\min}, t_{\max}]}$ is a geodesic in the Bures-Wasserstein sense, see Proposition 1, and

$$\min_{t \in [t_{\min}, t_{\max}]} BW_2^2(\pi(A + tX), \Sigma_i) = \min_{t \in [t_{\min}, t_{\max}]} \inf_{Q_i \in SO_d} \|A + tX - \Sigma_i^{1/2}Q_i\|^2$$

$$= \inf_{Q_i \in SO_d} \|A + p_{(A, X)}(t_i)X - \Sigma_i^{1/2}Q_i\|^2,$$

where $t_i = \langle \Sigma_i^{1/2}Q_i - A, X \rangle$ is the (orthogonal) projection time of $\Sigma_i^{1/2}Q_i$ onto the line $t \mapsto A + tX$.

We therefore deduce that a set of solution $(A, X, (Q_i)_{i=1}^n)$ of equation 12 defines a proper geodesic $(\pi(A + tX))_{t \in [t_{\min}, t_{\max}]}$, solution of problem in equation 11.

$\square$

**Proposition 13** (Proposition 5 in main). *Let $\nu_i = \mathcal{N}(m_i, \sigma_i^2)$ for $i = 1, \ldots n$ be $n$ univariate Gaussian distributions. The first principal geodesic component $t \in [0, 1] \mapsto \mu(t)$ solving equation 1 remains in the geodesic space of Gaussian distributions for all $t \in [0, 1]$.*

*Proof.* Let $\mathrm{Prob}_2(\mathbb{R})$ be the set of a.c. probability measures on $\mathbb{R}$ that have finite second moment, and $\mathcal{Q}$ the set of corresponding quantile functions :

$$\mathcal{Q} = \{F_\nu^{-1}; \ \nu \in \mathrm{Prob}_2(\mathbb{R})\}$$

$\mathcal{Q}$ is the set of increasing, left-continuous functions $q : (0, 1) \to \mathbb{R}$, and a convex cone in $L^2([0, 1])$, the set of square-integrable functions on $[0, 1]$. The mapping

$$\Phi\colon \nu \mapsto F_\nu^{-1} \tag{33}$$

defines an isometry between $\mathrm{Prob}_2(\mathbb{R})$ equipped with the Wasserstein metric, and $\mathcal{Q}$ equipped with the $L^2$ metric (see e.g. Bigot et al. (2017)), that is, for any $\mu, \nu \in \mathrm{Prob}_2(\mathbb{R})$,

$$W_2(\mu, \nu) = \|F_\mu^{-1} - F_\nu^{-1}\|_{L^2([0, 1])}.$$

The map $\Phi$ in equation 33 also defines an isometry from the set of (univariate) Gaussian distributions to the set of all Gaussian quantile functions $\mathcal{G}$. This space $\mathcal{G}$ is the upper-half of the plane $\mathcal{F}$ spanned by the constant function $\mathbf{1}$ and the quantile function $F_0^{-1}$ of the standard normal distribution:

$$\mathcal{G} = \mathbb{R} \cdot \mathbf{1} + \mathbb{R}_+^* \cdot F_0^{-1} \subset \mathcal{F}\colon \ = \mathrm{span}(\mathbf{1}, F_0^{-1}).$$

Now, consider $n$ normal distributions $\nu_1, \ldots, \nu_n$, and $(\mu(t))_{t \in [0, 1]}$ the first principal geodesic component found by minimizing equation 1, the sum of squared residuals in $\mathrm{Prob}_2(\mathbb{R})$. Since $\mu$ is a Wasserstein geodesic in $\mathrm{Prob}_2(\mathbb{R})$ and $\Phi$ is an isometry, the curve $t \mapsto \Phi(\mu)(t) = F_{\mu(t)}^{-1}$ is an $L^2([0, 1])$-geodesic in $\mathcal{Q}$, i.e. a line segment

$$t \in [0, 1] \mapsto F_{\mu(t)}^{-1} = (1 - t)F_{\mu(0)}^{-1} + tF_{\mu(1)}^{-1}.$$

Since $\{\mathbf{1}, F_0^{-1}\}$ forms an orthonormal basis of $\mathcal{F}$, the orthogonal projection of this line segment on $\mathcal{F}$ is given by

$$t \in [0, 1] \mapsto \langle F_{\mu(t)}^{-1}, \mathbf{1}\rangle\mathbf{1} + \langle F_{\mu(t)}^{-1}, F_0^{-1}\rangle F_0^{-1},$$

which lies in $\mathcal{G}$. To see this, we need to show that the following value is positive:

$$\langle F_{\mu(t)}^{-1}, F_0^{-1}\rangle = \int_0^1 F_{\mu(t)}^{-1}(y)F_0^{-1}(y)dy = \int_{\mathbb{R}} xF_0^{-1} \circ F_{\mu(t)}(x)d\mu(t)(x) = \mathbb{E}(XT(X)),$$

where $X \sim \mu(t)$ and $T = F_0^{-1} \circ F_{\mu(t)}$ is the Monge map from $\mu(t)$ to the standard normal distribution. Since $T$ is increasing, we indeed have $\mathbb{E}(XT(X)) > 0$ (see e.g. the proof of Theorem 2.2 in Schmidt (2014)).

Finally, since $\Phi(\mu)$ orthogonally projects from $\mathcal{Q}$ to $\mathcal{G}$ w.r.t the $L^2$ metric and $\Phi$ defines an isometry, we get that the geodesic $\mu$ orthogonally projects to a geodesic $\pi(\mu)$ in the space of Gaussian distributions, w.r.t. the Wasserstein metric. By the distance minimizing property of orthogonal projections, we know that the cost function in equation 1 evaluated at $\pi(\mu)$ is no larger than its value at $\mu$. Since $\mu$ is optimal, we get that $\mu = \pi(\mu)$ and $\mu$ belongs to the space of Gaussian distributions. $\qquad\square$

**Proposition 14.** *Let $\Sigma_1, \Sigma_2$ two SPD matrices that are diagonalizable in the same orthonormal basis, i.e.*

$$\Sigma_1 = P\begin{pmatrix} a_1^2 & 0 \\ 0 & b_1^2 \end{pmatrix} P^\top \quad and \quad \Sigma_2 = P\begin{pmatrix} a_2^2 & 0 \\ 0 & b_2^2 \end{pmatrix} P^\top,$$

*where $P$ is orthogonal. Then $BW_2^2(\Sigma_1, \Sigma_2) = (a_1-a_2)^2+(b_1-b_2)^2$, and thus the Bures-Wasserstein geodesic between $\Sigma_1$ and $\Sigma_2$ is given by*

$$\Sigma(t) = P\begin{pmatrix} ((1-t)a_1 + tb_1)^2 & 0 \\ 0 & ((1-t)a_2 + tb_2)^2 \end{pmatrix} P^\top, \quad 0 \le t \le 1.$$

*Proof.* This is a straightforward computation using equation 3. $\qquad\square$

**Proposition 15.** *Let us consider $n = 2p$ covariance matrices $\Sigma_i = \Sigma(a, b, \theta_i)$ as defined in equation 16, where $\theta_i = i\pi/n$ for $i = 0, \dots, n-1$. Then, the Bures-Wasserstein barycenter in equation 27 of these covariance matrices is given by $\bar\Sigma = (a+b)^2/4\,I$.*

*Proof.* Each pair of covariance matrices

$$\Sigma_i = P_{\theta_i}\begin{pmatrix} a^2 & 0 \\ 0 & b^2 \end{pmatrix} P_{\theta_i}^\top, \quad and \quad \Sigma_{i+p} = P_{\theta_i+\pi/2}DP_{\theta_i+\pi/2}^\top = P_{\theta_i}\begin{pmatrix} b^2 & 0 \\ 0 & a^2 \end{pmatrix} P_{\theta_i}^\top$$

are diagonalizable in the same basis, and so by Proposition 14, the geodesic from $\Sigma_i$ to $\Sigma_{i+p}$ is

$$\Sigma(t) = P_{\theta_i}\begin{pmatrix} ((1-t)a + tb)^2 & 0 \\ 0 & ((1-t)b + ta)^2 \end{pmatrix} P_{\theta_i}^\top, \quad 0 \le t \le 1.$$

In particular, the Fréchet mean is given by $\bar\Sigma = \Sigma(1/2) = ((a+b)/2)^2I$. Since each pair of covariance matrices has the same Fréchet mean, the Fréchet mean of the whole set $\Sigma_1, \dots, \Sigma_n$ is also given by $\bar\Sigma$. $\qquad\square$

**Proposition 16** (Proposition 4 in main). *Let $\Sigma \in S_2^{++}$ with eigenvalues $a^2, b^2$ and $\Sigma' = P_\theta\Sigma P_\theta^\top$ where $P_\theta$ is the rotation matrix of angle $\theta$. Then, denoting $\bar\Sigma = ((a+b)/2)^2 I$ we have*

$$\frac{BW_2^2(\Sigma, \Sigma')}{BW_{2,\bar\Sigma}^2(\Sigma, \Sigma')} = 1 - \left(\frac{a-b}{a+b}\right)^2 \cos^2\theta + O((a-b)^4). \tag{34}$$

*Proof.* Recall that the linearized Bures-Wasserstein distance at $\bar\Sigma$ between $\Sigma$ and $\Sigma'$ is given by the distance between their images by the Riemannian logarithm map $U := \mathrm{Log}_{\bar\Sigma}\Sigma$ and $U' := \mathrm{Log}_{\bar\Sigma}\Sigma'$ in the tangent space at $\bar\Sigma$, i.e.

$$BW_{2,\bar\Sigma}(\Sigma, \Sigma') = \|U - U'\|_{\bar\Sigma}^{BW},$$

where $\|\cdot\|^{BW}$ denotes the norm associated to the Bures-Wasserstein Riemannian metric in equation 18. As in any Riemannian manifold, the true geodesic distance can be approximated by this linearized distance in the tangent space, corrected by the curvature (see e.g. Lemma 1 in Harms et al. (2019)) :

$$BW_2^2(\Sigma, \Sigma') = \left(\|U - U'\|_{\bar{\Sigma}}^{BW}\right)^2 - \frac{1}{3}R_{\bar{\Sigma}}(U, U', U, U') + O(\|U\|_{\bar{\Sigma}}^{BW} + \|U'\|_{\bar{\Sigma}}^{BW})^6, \quad (35)$$

where $R_{\bar{\Sigma}}$ is the curvature tensor.

Recall from equation 18 that the Bures-Wasserstein norm of a vector $U$ is expressed in an eigenvector basis of the base point, here $\bar{\Sigma}$. Since any basis is an eigenvector basis of $\bar{\Sigma}$, it is convenient to choose that of $\Sigma$, which we can assume without loss of generality to be the canonical basis. Thus we write $\Sigma = D$ where $D = \mathrm{diag}(a^2, b^2)$ and $\Sigma' = P_\theta D P_\theta^\top$, and the norm associated to the Bures-Wasserstein Riemannian metric is given by

$$\|U\|_{\bar{\Sigma}}^{BW} = \frac{1}{2}\sum_{1 \leq i,j \leq 2}\frac{1}{d_i + d_j}U_{ij}^2$$

where the $d_i$'s are the eigenvalues of $\bar{\Sigma}$, given here by $d_1 = d_2 = ((a+b)/2)^2$. From Proposition 11 we have

$$U := \mathrm{Log}_{\bar{\Sigma}}\Sigma = (T - I)\bar{\Sigma} + \bar{\Sigma}(T - I),$$
$$U' := \mathrm{Log}_{\bar{\Sigma}}\Sigma' = (T' - I)\bar{\Sigma} + \bar{\Sigma}(T' - I),$$

where

$$T := \bar{\Sigma}^{-1/2}(\bar{\Sigma}^{1/2}\Sigma\bar{\Sigma}^{1/2})^{1/2}\bar{\Sigma}^{-1/2} = \frac{2}{a+b}D^{1/2},$$
$$T' := \bar{\Sigma}^{-1/2}(\bar{\Sigma}^{1/2}\Sigma'\bar{\Sigma}^{1/2})^{1/2}\bar{\Sigma}^{-1/2} = \frac{2}{a+b}P_\theta D^{1/2}P_\theta^\top,$$

and easily get

$$U = \frac{a^2 - b^2}{2}J, \quad U' = \frac{a^2 - b^2}{2}P_\theta J P_\theta^\top, \quad \text{where} \quad P_\theta J P_\theta^\top = \begin{pmatrix} \cos 2\theta & \sin 2\theta \\ \sin 2\theta & -\cos 2\theta \end{pmatrix}$$

and $J = \mathrm{diag}(1, -1)$. Thus after some computations we obtain

$$\|U\|_{\bar{\Sigma}}^{BW} = \|U'\|_{\bar{\Sigma}}^{BW} = |a - b|/\sqrt{2},$$
$$BW_{2,\bar{\Sigma}}(\Sigma, \Sigma') = \|U - U'\|_{\bar{\Sigma}}^{BW} = \sqrt{2}|(a - b)\sin\theta|. \quad (36)$$

To compute the curvature tensor, we use the following formula from (Thanwerdas, 2022, Table 4.7)

$$R_{\bar{\Sigma}}(U, U', U, U') = \frac{3}{2}\sum_{i,j}\frac{d_i d_j}{d_i + d_j}[U_0, U_0']_{ij}^2$$

where $[A, B] = AB - BA$ is the Lie bracket of matrices, $U_0$ and $U_0'$ are the only symmetric matrices verifying the Sylvester equations $U = U_0\bar{\Sigma} + \bar{\Sigma}U_0$ and $U' = U_0'\bar{\Sigma} + \bar{\Sigma}U_0'$ respectively. Since $\bar{\Sigma}$ is a multiple of the identity, we easily get

$$U_0 = \frac{a - b}{a + b}J, \quad U_0' = \frac{a - b}{a + b}P_\theta J P_\theta^\top$$

and straightforward computations yield

$$R_{\bar{\Sigma}}(U, U', U, U') = \frac{3}{2}\frac{(a - b)^4}{(a + b)^2}\sin^2 2\theta. \quad (37)$$

Finally, putting together equation 35, equation 36 and equation 37 and we obtain

$$BW_2^2(\Sigma, \Sigma') = BW_{2,\bar{\Sigma}}^2(\Sigma, \Sigma') - 2\frac{(a - b)^4}{(a + b)^2}\sin^2\theta\cos^2\theta + O((a - b)^6),$$

and dividing by the squared linearized optimal transport distance yields the desired result. $\square$

### D.2 Implementation of GPCA for Gaussian distributions

As described in Section 3, the first and second components of geodesic PCA are respectively found by solving the minimization problems in equation 12 and equation 13. The geodesic components are given by

$$\Sigma_i(t) = (A_i + tX_i)(A_i + tX_i)^\top, \quad \text{for} \quad i = 1, 2,$$

where $A_1 \in GL_d$ and $X_1 \in \text{Hor}_{A_1}$ are minimizers of equation 12, and $A_2 \in GL_d$ and $X_2 \in \text{Hor}_{A_2}$ minimizers of equation 13. The matrix $\pi(A_2)$ is the crossing point through which all geodesic components intersect, see Figure 2. The higher order components are found in a analogous way: for the $k$-th component, we search for a horizontal segment $t \mapsto A_k + tX_k$ where $A_k$ is set to the previous position in the fiber, $A_k = A_{k-1}$ (which implies that the horizontal segments parameterizing the geodesics in $\text{GL}_d$ intersect at the same point) and the horizontal velocity vector $X_k$ is orthogonal to the lifts of the velocity vectors of the previous component. Thus, the $k$-th component, $k \geq 3$, solves:

$$\begin{aligned} &\inf \ F(A_k, X_k, (Q_i)_{i=1}^n) \\ \text{subject to} \quad & A_k = A_{k-1}, \ X_k \in \text{Hor}_{A_k}, \ \|X_k\|^2 = 1, \\ & \langle X_k, X_{k-\ell} \rangle = 0, \ 1 \leq \ell \leq k-1, \ Q_1, \ldots, Q_n \in SO_d. \end{aligned} \quad (38)$$

Following Huckemann et al. (2010) and Calissano et al. (2024), we propose an iterative algorithm to implement these components, that, for each component, alternates two steps:

(Step 1) minimization of the objective function $F$ (see equation 12) with respect to $(Q_i)_{i=1}^n$ for fixed $(A, X)$,

(Step 2) minimization of the objective function $F$ with respect to $(A, X)$ for fixed $(Q_i)_{i=1}^n$.

In dimension $d = 2$, any rotation matrix $Q$ can be parametrized by a scalar angle $\theta$ and both steps are solved using the Sequential Least Squares Programming (SLSQP) algorithm (see e.g. Ma et al. (2024)) available on the *scipy python library* and given by Virtanen et al. (2020). In higher dimension, each minimization with respect to a rotation matrix is performed using Riemannian gradient descent on $SO_d$, relying on the Riemannian geometry of $SO_d$ induced by the standard Frobenius metric of the ambient space $\mathbb{R}^{d \times d}$. In particular we use the exponential map implemented in the *Python library geomstats* developed by Miolane et al. (2020). More details on the Riemannian geometry of $SO_d$ and the Riemannian gradient descent procedure can be found e.g. in (Boumal, 2023, Sections 7.4 and 4.3).

Unfortunately, we cannot ensure the convergence of the iterates of the proposed block alternating algorithm, as classical arguments require uniqueness of the minimizer at each iterations as proven in Powell (1973). This is unachievable in our problem: the line with base point $A$ and direction $X \in \text{Hor}_A$ and the line with base point $AQ$ and direction $XQ \in \text{Hor}_{AQ}$ for $Q \in O_d$ project onto the same geodesic in the bottom space. However, regarding (Step 1), and thanks to Theorem 3.7 in Huang & Wei (2022), we have for fixed $(A, X)$ that the cost function $f: (Q_1, \ldots, Q_n) \mapsto F(A, X, (Q_i)_{i=1}^n)$ has the Riemannian Kurdyka-Lojasiewicz property at any point of $(O_d)^n$. Finally, we have the convergence of the iterates towards an accumulation point thanks to Theorem 3.14 in Zhou et al. (2024). The three assumptions in this theorem are verified in our case : Assumption (3.5) ($L$-Retraction Smoothness) is obtained because $\text{grad} f$ is Lipschitz, and Corollary 10.54 in Boumal (2023); Assumption (3.7) (bounded from below) directly holds because $f \geq 0$; Assumption (3.8) (ndividual Retraction Lipschitzness) is verified thanks to Corollary 10.47 in Boumal (2023).

**Scalability of the algorithm** Surely, the computational time of our algorithm for Gaussian distributions will increase with the dimension. However, the algorithm can be made less sensitive to the number of input covariance matrices by parallelizing (Step 2) of our algorithm, which consists in updating the orthogonal matrices $(Q_i)_{i=1}^n$. This would significantly reduce the overall computational cost of the algorithm. Also, we currently use the scipy toolbox to solve (Step 1), which could also be accelerated using a more powerful optimization toolbox.

# E  Hyperparameters

## E.1  Hyperparameters setting

| Hyperparameter | Value |
|---|---|
| $f_\psi$ architecture | dense MLP
$d \to 128 \to 128 \to 128 \to 128 \to 1$
ELU activation functions |
| $f_\psi$ optimizer | Adam
step size $= 0.0005$
$\beta_1 = 0.9$
$\beta_2 = 0.999$ |
| $\varphi_\theta$ architecture | dense MLP
$d \to 128 \to 128 \to 128 \to 128 \to d$
RELU activation functions |
| $\varphi_\theta$ optimizer | Adam
step size $= 0.0005$
$\beta_1 = 0.9$
$\beta_2 = 0.999$ |
| $t_i$ optimizer | Adam
step size $= 0.005$
$\beta_1 = 0.9$
$\beta_2 = 0.999$ |
| batch size | 1024 |
| number of gradient steps first component | 120,000 |
| number of gradient steps second component | 200,000 |
| $\lambda_\mathcal{O}$ | 1.0 |
| $\lambda_\mathcal{I}$ | 1.0 |

Table 1: Hyperparameters used across all experiments.

All experiments were conducted on a single V100 GPU with 32GB of memory, using a shared set of hyperparameters detailed in Table 1. The same hyperparameters are used for computing both the first and second geodesic components, except for the number of gradient steps (see Table 1), which is increased for the second component. This is likely due to the additional complexity introduced by the intersection and orthogonality constraints enforced through regularization. Both $f_\psi$ and $\varphi_\theta$ are implemented as standard multilayer perceptrons (MLPs) with four hidden layers of width 128. We use ELU activation functions in $f_\psi$ because its gradient is used to parameterize a transport map in our formulation, and ELUs are commonly employed in such settings. The Sinkhorn divergence $S_\varepsilon$ is used in the loss function as a surrogate for the squared Wasserstein distance to compute the geodesic components. The regularization parameter $\varepsilon$ must be adapted to the scale of the data; we set it as $\varepsilon = 0.01 \, \mathbb{E}_{x,x' \sim \nu_i} \|x - x'\|^2$, where the expectation is approximated via Monte Carlo using the current minibatch samples. Note that setting $\varepsilon$ this way is the default configuration in the OTT-JAX library. For computing the second geodesic component, we fix the regularization coefficients $\lambda_\mathcal{O}$ and $\lambda_\mathcal{I}$ to 1.0, which we found to be robust across all experiments. While increasing them (e.g., to 10.0) typically yields similar results, excessively large values may degrade performance. Conversely, if these regularization terms are too small, the algorithm tends to recover the first component as the second, due to its lower cost. In practice, we monitor the regularization terms during optimization to ensure they decrease sufficiently relative to their initial values, confirming that the optimization effectively optimize the intersection and orthogonality constraints. To determine the hyperparameters in Table 1, we performed a grid search over the optimizer learning rate for the $t_i$ in $5e^{-4}, 1e^{-3}, 5e^{-3}, 1e^{-2}$, and over the regularization coefficients $\lambda_\mathcal{O}$ and $\lambda_\mathcal{I}$ in $0.1, 1.0, 10.0, 100.0$. We found that setting both regularization terms to 1.0 consistently yielded good performance across all experiments, see Section E.2.

**Note on $\varphi$ parameterization.** Note that although $\varphi$ is theoretically required to be a diffeomorphism in Otto's parameterization of geodesics (equation 9), we parameterize it using a simple MLP. Initially, we experimented with normalizing flows to ensure invertibility, but observed that a standard MLP yielded similar results. In Otto's geodesic framework, $\varphi$ serves to modify the reference measure $\rho$ and define the measure at $t = 0$ along the geodesic. If $\varphi$ is not a diffeomorphism and the pushforward $\varphi_{\#}\rho$ is not absolutely continuous, the resulting geodesic becomes degenerate, which may hinder optimization of the loss equation in equation 1. In practice, however, we found that the MLP $\varphi_\theta$ reliably produces absolutely continuous measures, which is sufficient for our method.

## E.2 IMPACT OF THE REGULARIZATIONS ON GPCA

For the estimation of the second GPCA component, we introduce two regularization terms, $\mathcal{I}(\xi_{\theta,\psi}, \xi_{\theta_2,\psi_2}, t^1_{\text{inter}}, t^2_{\text{inter}})$ and $\mathcal{O}(\nabla f_\psi(\varphi_\theta), \nabla f_{\psi_2}(\varphi_{\theta_2}))$, with their associated regularization coefficients $\lambda_I$ and $\lambda_O$. The first term enforces that the two components intersect, while the second ensures that the components remain orthogonal. Experimentally, we observe that setting both coefficients to $\lambda_I = \lambda_O = 1.0$ robustly enforces these constraints across all experiments while still producing meaningful principal components. Conversely, if these regularization terms are too small, the algorithm tends to recover the first component as the second, at it gives the lowest cost. In practice, we monitor the regularization terms during optimization to ensure they decrease sufficiently relative to their initial values. This permits to confirm that the optimization effectively optimize the intersection and orthogonality constraints. This section aims at quantifying the impact of the two regularizing coefficients $\lambda_I$ and $\lambda_O$ on the computed geodesics. We focus on the 3D point-cloud experiments with lamps.

### E.2.1 ORTHOGONALITY REGULARIZATION

In this part, we set the regularization term $\lambda_I$ to 1.0 and compute GPCA for different values of $\lambda_O$. The resulting second component is shown in Figure 20. The GPCA cost of this component, as defined in equation 15, together with the quantity measuring the orthogonality between components, $\mathcal{O}(\nabla f_\psi(\varphi_\theta), \nabla f_{\psi_2}(\varphi_{\theta_2}))$, are reported in Table 2. The quantities reported in Table 2 are estimated on batches of size 2048. The variance is computed over 100 runs for the orthogonality measure and 5 runs for the GPCA cost. Note that each run of the orthogonality estimation already involves computing 100 Wasserstein distances, since we have 100 point clouds.

Note that the GPCA cost of the second component should be compared with that of the first component, which is around 3.0. Table 2 shows that for low values of $\lambda_O$ (i.e., 0.001 and 0.01), the orthogonality quantity is large, and the recovered "second" component is in fact identical to the first component, as illustrated in Figure 20. This is also reflected in the GPCA cost (see Table 2), which matches the one of the first component. For higher values of $\lambda_O$ (0.1, 1.0, 10.0, 100.0), the algorithm successfully recovers a distinct second component. For the highest value (i.e., $\lambda_O = 100.0$), a loss of performance is observed.

| $\lambda_O$ | Orthogonality: $\mathcal{O}(\nabla f_\psi(\varphi_\theta), \nabla f_{\psi_2}(\varphi_{\theta_2}))$ | GPCA cost (second component) |
|---|---|---|
| 0.001 | $0.909 \pm 0.005$ | $2.96 \pm 0.01$ |
| 0.01 | $0.811 \pm 0.008$ | $3.00 \pm 0.01$ |
| 0.1 | $2.1 \times 10^{-3} \pm 3 \times 10^{-4}$ | $5.75 \pm 0.02$ |
| 1.0 | $3.1 \times 10^{-4} \pm 6.5 \times 10^{-5}$ | $5.76 \pm 0.02$ |
| 10.0 | $2.2 \times 10^{-4} \pm 5 \times 10^{-5}$ | $5.89 \pm 0.02$ |
| 100.0 | $1.1 \times 10^{-5} \pm 2 \times 10^{-6}$ | $5.99 \pm 0.02$ |

Table 2: Orthogonality regularization value and second-component loss for different values of $\lambda_O$.

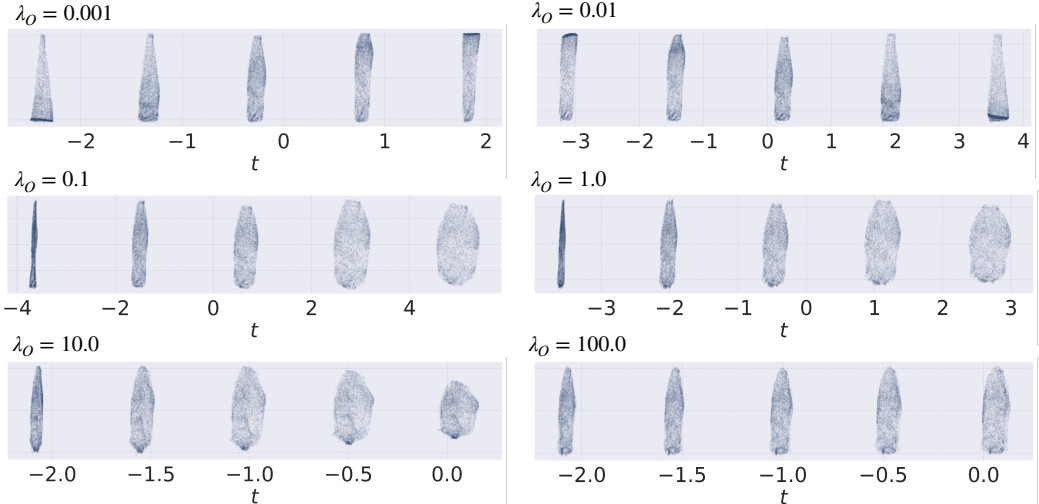

Figure 20: Empirical distributions sampled uniformly along the geodesics associated with the second GPCA principal component for different values of the regularization coefficient $\lambda_O$. In all experiments, the other regularization coefficient is fixed at $\lambda_I = 1.0$.

### E.2.2 REGULARIZATION ON THE INTERSECTION OF THE GEODESICS

In this part, we set the regularization term $\lambda_O$ to 1.0 and compute GPCA for different values of $\lambda_I$. The second component is displayed in Figure 21; the GPCA cost of this component, as well as the quantity measuring the intersection of the components, $\mathcal{I}(\xi_{\theta,\psi}, \xi_{\theta_2,\psi_2}, t_{\text{inter}}^1, t_{\text{inter}}^2)$, are reported in Table 3. The quantities reported in Table 3 are estimated on batches of size 2048. The variance is computed over 100 runs for the intersection measure and 5 runs for the GPCA cost.

We observe from the recovered geodesics in Figure 21 that this regularization term plays a less significant role than the orthogonality term. Moreover, Table 3 shows that increasing $\lambda_I$ does not affect negatively the GPCA cost of the recovered component.

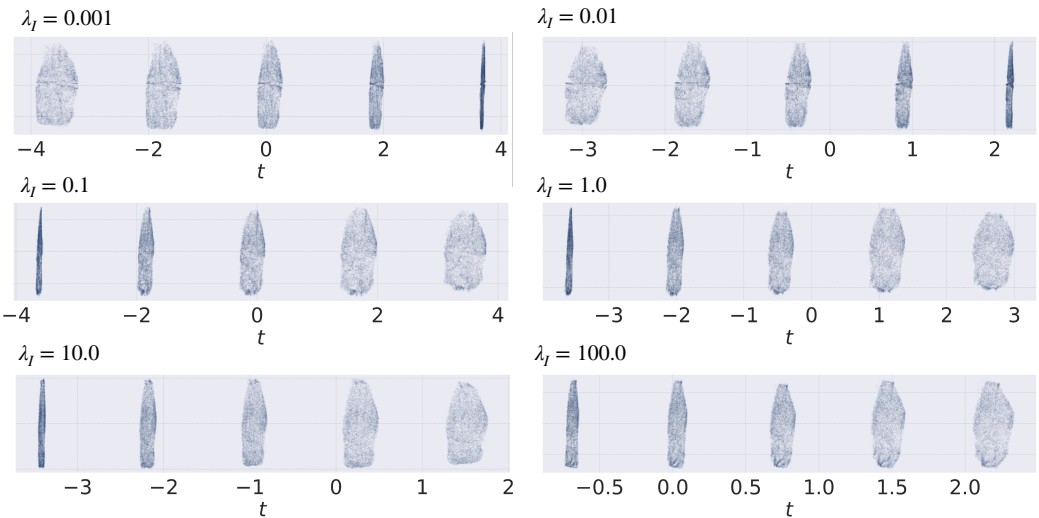

Figure 21: Empirical distributions sampled uniformly along the geodesics associated with the second GPCA principal component for different values of the regularization coefficient $\lambda_I$. In all experiments, the other regularization coefficient is fixed at $\lambda_O = 1.0$.

| $\lambda_I$ | Intersection: $\mathcal{I}(\xi_{\theta,\psi}, \xi_{\theta_2,\psi_2}, t_{\text{inter}}^1, t_{\text{inter}}^2)$ | GPCA cost (second component) |
|---|---|---|
| 0.001 | $6.5 \times 10^{-2} \pm 1 \times 10^{-3}$ | $5.76 \pm 0.02$ |
| 0.01 | $2.3 \times 10^{-2} \pm 4 \times 10^{-4}$ | $5.77 \pm 0.02$ |
| 0.1 | $2.7 \times 10^{-3} \pm 1 \times 10^{-4}$ | $5.77 \pm 0.01$ |
| 1.0 | $1.0 \times 10^{-3} \pm 6 \times 10^{-5}$ | $5.76 \pm 0.02$ |
| 10.0 | $7.2 \times 10^{-5} \pm 3 \times 10^{-6}$ | $5.74 \pm 0.02$ |
| 100.0 | $3.1 \times 10^{-5} \pm 1 \times 10^{-6}$ | $5.91 \pm 0.02$ |

Table 3: Squared Euclidean distance between $\xi_1(t_{\text{inter}}^1)$ and $\xi_2(t_{\text{inter}}^2)$ and second-component loss for different values of $\lambda_I$.

### E.2.3 SCALABILITY OF OUR GPCAGEN ALGORITHM

For general distributions, there are two types of "scaling" that can affect the algorithm:

1. Number of probability measures ($n$): The number of measures $\nu_i$ directly determines the iterations of the inner loop in Algorithm 1 (line 3). Consequently, the training time scales linearly with $n$.

2. Dimension of the space ($d$): As the dimension of the space in which the $\nu_i$ lies increases, the main challenge consists in accurately estimating the maximum and minimum eigenvalues that the Hessian of $f$ can take. As discussed with reviewer oUMT, in high dimensions, it becomes necessary to use algorithms that avoid computing the full Hessian and instead rely on matrix-vector products, such as the LOBPCG algorithm Duersch et al. (2018). Furthermore, rather than relying solely on the samples in the training batch, an adversarial approach would be needed to track the eigenvectors corresponding to the worst-case eigenvalues.

## F USE OF LARGE LANGUAGE MODELS (LLMS)

LLMs were used only to assist with polishing the writing; all research ideas, experiments, and analyses were conducted independently by the authors.

