# OpenReview forum: "On the Wasserstein Geodesic Principal Component Analysis of probability measures"
_ICLR.cc/2026/Conference — ICLR 2026 Oral_

### Official Review · Reviewer_Pw4V · 2025-10-26

**Soundness:** 3
**Presentation:** 3
**Contribution:** 3
**Rating:** 4
**Confidence:** 3

**Summary:**

This paper concern with the problem of Geodesic Principal Component Analysis (GPCA), i.e., given $n$ probability measure $\nu_1,\ldots,\nu_n$, we gradually construct a sequence of geodesics $\mu_1,\ldots,\mu_k$, known as the principle components, such that
\begin{equation*}
    \mu_i = \inf_{t \mapsto \mu(t) \text{ geodesics}} \sum_{i=1}^n \inf_{t_i} W_2^2(\mu(t_i),\nu_i),
\end{equation*}
and the subsequent principle components satisfy the same cost function with additional constrains about orthogonality with previous principle components. To resolve this problem, the authors leverage tool from Otto-Wassernstein geometry. They propose two algorithms to tackle two different settings: centered Gaussian distributions and a.c. probability measures: for first setting, the problem reduces to the matrix quadratic optimisation, for the second setting, by choosing a Gaussian reference measure, the problem reduces to optimise a function with neural network. Then, they validate their algorithms using toy examples (for Gaussian distributions) and some real dataset (MNIST, 3D point cloud, etc. )

**Strengths:**

$\bullet$ Originality: The problem appears to be novel and applicable to the real dataset, as the authors demonstrate in their experience.

$\bullet$ Quality: Firstly, the authors provide both theoretical intuition and experimental verification for the algorithm.

$\bullet$ Clarity: The article is easy to follow in general.

$\bullet$ Significance: The algorithm provide a method to analysis the principle components of data. In the real data, they relate to some important aspects of data, such as brightness, shape, etc.

**Weaknesses:**

I think that the article does not provide a quite clear theoretical verification: despite of the fact that the authors provide a nice intuition from Otto-Wassernstein geometry, they do not provide the proof of convergence of these two algorithms. For example, in Lemma 1, they just point out that the loss function admit one global minimum and do not show that the algorithm converge to the, at least, stationary point. In addition, for the case of centered Gaussian distributions, as the author admit, it is unknown that the optimal solution is still Gaussian distribution in general distribution space. Another aspect is that some of the theoretical results seems to not useful in the analysis of these two algorithms. For example, I am not sure that how we use the proposition 6, 7, 8 in our algorithms.

Minor suggestion:
- Line 32: accound → account

- Equation 26: inconsistent notation, the author use "]" to denote open bracket "(" and "[" to denote close bracket, while in other formula they use the "(" and ")".

- "hessian" should be "Hessian"

- Line 180: GL should be italic

- The proof for coercivity is not clear (among the notion of coercivity, which one that you use, and why the norm $\|()\|$ goes to infinity implies the coercivity).

**Questions:**

1. As pointed out in the weakness section, I miss the point that how can we
use the proposition 6, 7, 8 in our algorithms.

2. Can we use this algorithms for some LLMs model or other complicated image tasks?

3. Can you point out an example that the global minimum point in Lemma 1 is not unique?

---

> ### Author Response · Authors · 2025-11-22
>
> Dear Reviewer,
>
> Thank you for taking the time to review our paper and for your comments and suggestions. We have tried to address them below.
>
> **I think that the article does not provide a quite clear theoretical verification: despite of the fact that the authors provide a nice intuition from Otto-Wassernstein geometry, they do not provide the proof of convergence of these two algorithms. For example, in Lemma 1, they just point out that the loss function admit one global minimum and do not show that the algorithm converge to the, at least, stationary point.**
>
> For the Gaussian distributions case, it is very difficult to prove the convergence of the iterates for a block alternating algorithm, especially when each block corresponds to a constrained optimization problem. Still, our algorithm provides a monotone sequence with respect to the objective functional F. As an example, Bertsekas in [R4] showed that if the minimum with respect to each block is unique, then any accumulation point of the sequence generated by the method is also a stationary point, implying that obtaining a converging sequence is in general hard when having non-uniqueness for each block, which is our case. Additionally, we believe that proving that the proposed optimization problem using Otto-Wasserstein geometry solves the GPCA problem is an interesting contribution.
>
> For the a.c. case, the use of neural networks makes it possible to compute GPCA but prevents us from obtaining guarantees on the convergence of the algorithm. In fact, the minimum of the loss function in Equation (15) corresponds to the GPCA components, but we cannot guarantee the convergence of the algorithm.
>
> [R4] Bertsekas, Nonlinear Programming (1999)
>
> **In addition, for the case of centered Gaussian distributions, as the author admit, it is unknown that the optimal solution is still Gaussian distribution in general distribution space.**
>
> Indeed, in dimension greater than one, it is not clear if the GPCA problem that we consider in the space of centered Gaussian distributions is equivalent to the GPCA problem in the general Wasserstein space when applied to a set of Gaussian distributions. More generally, it is an interesting open question whether, when performing GPCA in a totally geodesic submanifold, the components remain in the submanifold.
>
> However, the Bures-Wasserstein is a metric that has raised a lot of interest in the past years as shown by the literature, and it is also related to other fields such as quantum information, see Bhatia et al. (2019). Thus we believe that GPCA with respect to that metric is interesting in itself. Furthermore, in some applications it may be important that the geodesic components found by PCA are Gaussian, e.g. for interpretability purposes or to remain in the Gaussian representation space chosen for the data, a strategy chosen e.g. in [R5] for Gaussian mixtures.
>
> [R5] J. Delon and A. Desolneux, A Wasserstein-type distance in the space of Gaussian mixture models. SIAM Journal on Imaging Sciences, 13(2), 936-970, 2020.
>
> **Another aspect is that some of the theoretical results seems to not useful in the analysis of these two algorithms. For example, I am not sure that how we use the proposition 6, 7, 8 in our algorithms.**
>
> Proposition 6, 7 and 8 are important for the parametrization of geodesic components used in the GPCA algorithm in the general Wasserstein space (Algorithm 1). Prop. 6 states that horizontal vectors are gradients of functions, and Prop. 7 states that horizontal straight lines in the space of diffeomorphisms project to Wasserstein geodesics in the space of measures. They both imply that Wasserstein geodesics can be parametrized using Equation (17) (in Prop. 8), which is the parametrization that we use in the GPCAgen algorithm. In particular, it allows us to define the loss function of the algorithm in Equation (15). Moreover, these results also show that the proper notion of orthogonality for the Wasserstein metric is through the $L^2$-inner product with respect to the reference measure $\rho$ (the standard Gaussian distribution), which we use in the regularization terms $\mathcal I$ and $\mathcal O$ (see Section 4 of the paper).
>
> **Minor suggestions**
>
> Thank you for correcting typos and for your suggestions, we took them into account in our revised version.
>
> **The proof for coercivity is not clear (among the notion of coercivity, which one that you use, and why the norm |()|  goes to infinity implies the coercivity).**
>
> The definition of coercivity that we use is the following : $f : X\to \bar{\mathbb{R}}$ is coercive if $\underset{\Vert x\Vert\to\infty}{\lim}\ f(x) = \infty$ (see e.g. [R6]). In our paper, we demonstrate that the function F in the proof of Lemma 1 goes to infinity when the norm of the triplets $(A,X,Q_i)$ goes to infinity.
>
> [R6] Zalinescu, Convex analysis in general vector spaces. World scientific, 2002

---

> > ### Author Response · Authors · 2025-11-22
> >
> > **Can we use this algorithms for some LLMs model or other complicated image tasks?**
> >
> > Practical use cases of our method include, for example, classification and clustering following GPCA (as illustrated in Figure 12), as well as outlier detection (see the newly added Section A.3 in the Appendix). GPCA is useful for data viewed as samples from a distribution, where no natural ordering exists and the points cannot be represented as a fixed-dimensional vector, as is the case for 3D point clouds.
> >
> > **Can you point out an example that the global minimum point in Lemma 1 is not unique?**
> >
> > In Figure 8, we present an example of GPCA for a set of matrices with same eigenvalues.  We obtain two different solutions for the first and second geodesic components in this case (Figure 8, middle and right).

---

> > > ### Comment · Reviewer_Pw4V · 2025-11-22
> > >
> > > I have carefully read the comment of authors. Despite of the fact that from my viewpoint, the theoretical contribution of this article is somewhat limited, I admit that the theoretical aspect here is quite hard to implement a rigorous proof. The authors have almost addressed all of my concern in the comment. Due to the significant algorithmic contribution, I decided to raise the score to 6.

---

> > > > ### Author Response · Authors · 2025-11-24
> > > >
> > > > Dear Reviewer,
> > > >
> > > > Thank you for taking the time to read our rebuttal. We are glad to have addressed your questions and are thankful for your score increase.

---

### Official Review · Reviewer_F5cW · 2025-10-31

**Soundness:** 4
**Presentation:** 4
**Contribution:** 3
**Rating:** 8
**Confidence:** 4

**Summary:**

The paper introduces a framework for Geodesic Principal Component Analysis (GPCA) of probability measures under the Otto-Wasserstein geometry. It provides exact solutions for Gaussian distributions by lifting computations to the space of invertible matrices and extends the method to general absolutely continuous measures using a neural network-based parametrization of Wasserstein geodesics.  Experiments on Gaussian data, 3D point clouds, and images show that captures meaningful nonlinear modes of variation and highlight the difference to tangent PCA.

**Strengths:**

**Exposition:**
I found the paper to be very well written. It has a common thread running through it that makes it easy to follow the story. Thus, I could read it in one go and understood all the core ideas. Furthermore, I think all the necessary information is included in the paper needed to reproduce the method and the experiments. The division of information between main text and appendix is also sensible.

**Novelty:**
I think the introduced method is novel and advances the state-of-the-art in measure-based (geometric) data analysis. Thus, I consider it to be a contribution worthy of being published at ICLR. The distinction to similar methods (esp. TPCA) is clearly made and well-explained.

**Theory:**
The included theory is sound and reasonable. To me, all the included theoretical results clearly connect to the overall story of the paper and are well explained.

**Reproducability:**
With the provided information, the method (esp. the ac. formulation) should be easy to implement. I liked especially that pseudo-code (Alg. 1) was provided, as this always makes things easier to understand. Furthermore, all the experiments are well documented to level where one could reproduce them.

**Weaknesses:**

**Scalability:** I am a bit worried about the scalability of the method. All examples are conducted on a small scale with at most two components. Thus, the paper leaves the gap what would happen for larger datasets and what kind of resources the method requires in such a scenario. It would be great if the authors could discuss this in the paper and also illuminate if it is, indeed, a problem.

**Practical applications:**
This ties into the second weakness I see with the paper: a lack of practical applicability. As mentioned above, the examples are small and restricted to synthetic examples. Thus, I do not see on which applications the paper could have an (immediate) impact. As I see this as a more of a theoretical/modelling work, I do not consider this as a major issue. However, it would be nice if the authors could include a discussion what kind of applications they envision their method could be useful for (down the line) and how they image their ideas could impact other machine learning work.

**Questions:**

- Intro: It would be nice to list some concrete examples on the shortcomings of using $L_2$ for probability densities
  - It would be nice to show some concrete examples for prop. 6 in terms of measures and vector fields
  - Problem 38: Why not a global optimization instead of the block optimization?
  - In the paper, one component is computed after another. Could one find multiple components at once? Is it equivalent? Especially for the neural network part, I do not think this should be hard to implement?
  - What happens for Gaussian distributions with varying centers? Is there still a nice reduced model?
  - Typesetting: please use \colon instead of : when introducing functions
  - Can geodesics be extended in the Otto formulation beyond t_min/max? E.g. by choosing a new horizontal vector?
  - Language: I believe the correct usage is "to lift to \[a space\]" instead of "to lift in \[a space\]"
  - Fig. 3 and others: I believe it would be more instructional to show the cone as a volumetric object. This might be harder a render, but maybe the authors could give it a try.

---

> ### Author Response · Authors · 2025-11-22
>
> Dear Reviewer,
>
> Thank you for taking the time to review our paper and for your positive feedback on our work. We did our best to answer some of your concerns / questions below.
>
> **Scalability: I am a bit worried about the scalability of the method. All examples are conducted on a small scale with at most two components. Thus, the paper leaves the gap what would happen for larger datasets and what kind of resources the method requires in such a scenario. It would be great if the authors could discuss this in the paper and also illuminate if it is, indeed, a problem.**
>
> Indeed, we only propose two and three dimensional experiments; below, we outline the potential challenges we may face when scaling our method, and potential solutions:
>
> - For Gaussian distributions, indeed the computational time will increase with the dimension. However, the algorithm can be made less sensitive to the number of input covariance matrices by parallelizing (Step 2) of our algorithm, which consists in updating the orthogonal matrices $(Q_i)_{i=1}^n$. This would significantly reduce the overall computational cost of the algorithm. Also, we currently use the scipy toolbox to solve (Step 1), which could also be accelerated using a more powerful optimization toolbox.
>
> - For general distributions, there are two types of “scaling” that can affect the algorithm:
>
>   - Number of probability measures ($n$): The number of measures $\nu_i$ directly determines the iterations of the inner loop in Algorithm 1 (line 3). Consequently, the training time scales linearly with $n$.
>
>   - Dimension of the space ($d$): As the dimension of the space in which the $\nu_i$ lies increases, the main challenge consists in accurately estimating the maximum and minimum eigenvalues that the Hessian of $f$ can take. As discussed with reviewer oUMT, in high dimensions, it becomes necessary to use algorithms that avoid computing the full Hessian and instead rely on matrix-vector products, such as the LOBPCG algorithm [R1]. Furthermore, rather than relying solely on the samples in the training batch, an adversarial approach would be needed to track the eigenvectors corresponding to the worst-case eigenvalues.
> Still, previous works that proposed an exact GPCA in the Wasserstein space, without relying on linearization, were restricted to the one-dimensional case. In this regard, we have already pushed the scalability of GPCA by conducting experiments on real 3D datasets.
> We have added these discussions in Section D.2 for the Gaussian case and in Section E.2.3 for the GPCAgen algorithm.
>
> [R1] Duersch, J.A., Shao, M., Yang, C., Gu, M.: A robust and efficient implementation of LOBPCG. SIAM J. Sci. Comput. 40(5), 655–676 (2018). https://doi.org/10.1137/17M1129830
>
> **Practical applications: This ties into the second weakness I see with the paper: a lack of practical applicability. As mentioned above, the examples are small and restricted to synthetic examples. Thus, I do not see on which applications the paper could have an (immediate) impact. As I see this as a more of a theoretical/modelling work, I do not consider this as a major issue. However, it would be nice if the authors could include a discussion what kind of applications they envision their method could be useful for (down the line) and how they image their ideas could impact other machine learning work.**
>
> Thank you for this suggestion. We have added a sentence in the conclusion mentioning the possible applications of GPCA. These include classification and clustering following GPCA (see the illustrative experiment in Figure 12), as well as outlier detection (see the newly added Section A.3 in the Appendix).
>
> **Questions:**
> **Intro: It would be nice to list some concrete examples on the shortcomings of using L2 for probability densities**
>
> Even in the simpler case of one dimensional probability distributions, L2 fails to properly capture the modes of variations of a set of densities. The geodesic components will live in L2, and not in the space of probability density distributions. This implies that the projection of a density onto the geodesic will most likely result in a non-positive function, which does not integrate to one. Therefore, the components will be particularly hard to interpret. We have added this comment in the revised version.
>
> **It would be nice to show some concrete examples for prop. 6 in terms of measures and vector fields**
>
> We emphasize that Proposition 6 is a classical result from Otto’s paper describing the Riemannian interpretation of the Wasserstein distance, and is included in the appendix of the paper for the sake of completeness. This is why it does not seem necessary to us to illustrate this theoretical result specifically in the paper.

---

> > ### Author Response · Authors · 2025-11-22
> >
> > **Problem 38: Why not a global optimization instead of the block optimization?**
> >
> > Problem (38) is a joint optimization over (1) rotation matrices and (2) a pair (A,X) of matrices linked by a constraint. In dimension 2, the orthogonal group is one-dimensional and the optimization can be performed jointly using the Scipy libraries. However we have found empirically that the block optimization performed slightly better. In higher dimensions, the algorithm performing joint optimization did not converge and we had to separate it into two alternating optimizations, where the first one is performed through gradient descent on $SO(n)$ endowed with a natural metric.
> >
> > **In the paper, one component is computed after another. Could one find multiple components at once? Is it equivalent? Especially for the neural network part, I do not think this should be hard to implement?**
> >
> > Computing multiple components at once corresponds to a different PCA algorithm, distinct from the GPCA algorithm presented in this paper. While in the Euclidean case, there is a unique notion of PCA leading to nested subspaces, in curved settings such as the Wasserstein space, there are multiple, non-equivalent notions of PCA. Computing multiple components at once would require a totally new approach for the Gaussian case, such as barycentric subspace analysis [R3]. However, you are correct in noting that the GPCAGen algorithm can be easily adapted to compute solutions for this alternative algorithm.
> >
> > [R3] X. Pennec, Barycentric Subspace Analysis on Manifolds, The Annals of Statistics 46(6), 2016. https://doi.org/10.1214/17-AOS1636.
> >
> > **What happens for Gaussian distributions with varying centers? Is there still a nice reduced model?**
> >
> > Unfortunately, we no longer have a quotient structure for Gaussians with varying means, and this structure is central to apply our GPCA method. Moreover, although the Wasserstein distance between two Gaussian distributions is the product of the Euclidean distance on means and the Bures-Wasserstein distance on covariances, one cannot perform GPCA on means and covariances separately. Indeed, the means of the input Gaussian distributions impact the optimality of the geodesic components : the mean and covariance along the geodesic at optimality are not independent.
> >
> > **Typesetting: please use \colon instead of : when introducing functions**
> >
> > Thank you, we have modified this in the revision.
> >
> > **Can geodesics be extended in the Otto formulation beyond t_min/max? E.g. by choosing a new horizontal vector?**
> >
> > The interval [t_min, t_max] is the maximal interval of definition of the geodesic, and both times indicate when a geodesic reaches the border of the space of distributions. One could indeed choose to extend the component in a different direction. This would result in a geodesic by piece.
> >
> > **Language: I believe the correct usage is "to lift to [a space]" instead of "to lift in [a space]"**
> >
> > Thank you for noticing, this has been corrected.
> >
> > **Fig. 3 and others: I believe it would be more instructional to show the cone as a volumetric object. This might be harder a render, but maybe the authors could give it a try.**
> >
> > We agree that the visualization of the geodesic components inside the cone is tricky. Unfortunately we have not been able to get a better rendering of the cone without partially hiding the geodesic components, which are what we want to emphasize in the Figure.

---

### Official Review · Reviewer_2t9m · 2025-11-01

**Soundness:** 3
**Presentation:** 3
**Contribution:** 3
**Rating:** 6
**Confidence:** 4

**Summary:**

This paper introduces an algorithm for geodesic PCA using optimal transport ideas. This is done by generalizing a certain formulation of Euclidean PCA, giving an optimization problem for the first principle component involving geodesics in Wasserstein space. Further principal components are defined using orthogonality. There are two versions presented: for Gaussians, and for more general distributions relying on neural network approximation. Evaluations are performed on synthetic Gaussian-type data, and point clouds.

**Strengths:**

I liked the following:
* **Interesting problem.** Generalizing PCA to spaces of probability measures seems to be a generically useful tool, since comparing distributions is a central task throughout machine learning which recurs in many situations.
* **Technically sound - especially for Gaussian distributions.** The approach involves the Bures-Wasserstein geometry and relationships between certain matrix groups, and makes it easier to see
* **Easily provides use cases beyond what the authors have illustrated.** One way one can think about PCA is that it provides a very crude form of manifold learning, where the manifold is a subset of the ambient Euclidean space. This can be used to define for instance an outlier score, by comparing distance between each datapoint and the PCA subspace. The authors do not consider this, but the fact that it is easily possible reveals the general value of the technique.

**Weaknesses:**

I am worried about the following:
* **Use of regularization in neural network objectives.** In particular, using regularization to enforce geometric constraints is much weaker than incorporating them as a hard constraint via a clever parametrization. In practice, I suspect the different directions do not end up orthogonal, and it would be helpful to quantify how much this is a problem in practice, and how sensitive it is to hyperparameter tuning. I did not see an experiment directly addressing this.
* **Experiments: Gaussian case should have at least *some* kind of non-synthetic evaluation.** Gaussian experiments are limited to toy datasets designed to highlight differences with ordinary PCA. While this is certainly necessary, it would have been better to also include some kind of non-synthetic datasets where the Gaussian mean and covariance is estimated empirically. For example, in continuous action imitation learning and RL, one often considers policies consisting of neural networks that learn a mean and covariance. In this domain, the authors' form of PCA could be used for things like outlier detection. I am not suggesting the authors necessarily need to add this specific example, more that it's important to have *something* non-synthetic for the Gaussian case since it's a central part of the paper.
* **Experiments: why is there a theorem in the experiments section?** Theory should be presented separately from evaluation, this part should be moved to some part in main body.
* **Experiments: Gaussian case is missing obvious baselines.** In 5.1, a natural baseline is to compare against is Euclidean PCA in the space of Cholesky coefficients of the covariance matrix. How would this produce different results compared to Euclidean PCA with respect to the SPD cone coordinates, or the proposed method?
* **Experiments: non-Gaussian case is missing obvious baselines.** For the point cloud data presented, the most obvious way I can think of to do PCA is take some kind of off-the-shelf model like PointNet, and perform PCA in its latent space, and present the results. While this is certainly a much heavier method, I think some kind of visual comparison needs to be included in the main body, as this is the single most obvious performance question that readers might have.
* **Experiments: no evaluation of downstream applications of PCA like outlier detection.** The authors do not evaluate how good the principal components produced are for any downstream purpose, instead they are just plotted to show their properties. I would like to see at least some kind of quantitative evaluation. For instance, this could be considered for outlier detection, to compare how well linear vs. the proposed geodesic PCA performs. Non-linear PCA has been studied for this specific purpose before, see for instance "Quadric Hypersurface Intersection for Manifold Learning in Feature Space" by Pavutnitskiy et al. which includes full experiments on this. Note that I am not suggesting the authors implement this specific downstream application of PCA, instead I think it is important to have at least some kind of evaluation of this kind.

**Questions:**

The overwhelming majority of my concerns with this paper concern the comprehensiveness of evaluations and specifically comparisons with various baselines like PCA on PointNet latent space. Please address those comparisons, or argue why they are not appropriate.

---

> ### Author Response · Authors · 2025-11-22
>
> Dear Reviewer,
> We are very grateful for your time spent reading our paper, as well as for your suggestions on baselines to compare our methods. We did our best to address your concerns and questions below.
>
> **Use of regularization in neural network objectives. In particular, using regularization to enforce geometric constraints is much weaker than incorporating them as a hard constraint via a clever parametrization. In practice, I suspect the different directions do not end up orthogonal, and it would be helpful to quantify how much this is a problem in practice, and how sensitive it is to hyperparameter tuning. I did not see an experiment directly addressing this.**
>
> Indeed, we have resorted to regularization to enforce our geometric constraints by lack of a way to enforce them as hard constraints. To the best of our knowledge, there is no method or neural-network architecture that can enforce the gradients of NN-parameterized functions to be orthogonal. That being said, we believe that relying on regularization to impose constraints remains quite common in machine learning; see, for example, the Wasserstein Autoencoder [R2], whose setting is closely related to ours. In that case, the latent codes are enforced to match a predefined prior distribution by introducing a penalty term based on the Wasserstein distance between the distribution of the latent codes and the prior.
>
> In the case of the MNIST experiment in Figure 9 (Appendix A.1), one can clearly affirm that the recovered geodesics are indeed orthogonal. For the other datasets, the empirical value of the orthogonality regularizer provides a quantitative assessment of the orthogonality of the recovered components. We recognize that the paper was missing an experiment illustrating the impact of the regularization coefficients $\lambda_I$ and $\lambda_O$ on the recovered second component and on the values of the intersection and orthogonality regularizations. Following your suggestion, **we have added this experiment in the appendix of the revised version (see Appendix E.2.1 and E.2.2 of the updated version of the paper).**
>
> [R2] Tolstikhin, I., , Bousquet, O., Gelly, S., Scholkopf, B.: Wasserstein auto-encoders. ICML 2018.
>
> **Experiments: Gaussian case should have at least some kind of non-synthetic evaluation. Gaussian experiments are limited to toy datasets designed to highlight differences with ordinary PCA. While this is certainly necessary, it would have been better to also include some kind of non-synthetic datasets where the Gaussian mean and covariance is estimated empirically. For example, in continuous action imitation learning and RL, one often considers policies consisting of neural networks that learn a mean and covariance. In this domain, the authors' form of PCA could be used for things like outlier detection. I am not suggesting the authors necessarily need to add this specific example, more that it's important to have something non-synthetic for the Gaussian case since it's a central part of the paper.**
>
> We conducted several experiments to demonstrate the usefulness of PCA on Gaussian distributions, approximating real dataset with their mean vector and covariance matrix. As the results obtained with TCPA and geodesic PCA were very similar, we did not consider that they provided additional knowledge and insights into the differences between these two techniques. That said, in general, it is interesting to add experiments solely to show the advantages of representing a dataset using a PCA technique, **which we have done in the revision (see Section 5.1) by  considering the Weather CORGIS Dataset to illustrate GPCA based on empirical covariance matrices.**
>
> **Experiments: why is there a theorem in the experiments section? Theory should be presented separately from evaluation, this part should be moved to some part in main body.**
> We had added Proposition 5 in the experiment section as it concerns a particular case that we illustrate directly after the proposition. Following your suggestion, we have moved Proposition 5 to the theoretical Section 3 on Gaussian distributions in the revised version.

---

> > ### Comment · Reviewer_2t9m · 2025-11-25
> >
> > Thanks very much for adding these extra checks. This significantly helps with my concerns about the paper. I have bumped my score.

---

> ### Author Response · Authors · 2025-11-22
>
> **Experiments: Gaussian case is missing obvious baselines. In 5.1, a natural baseline is to compare against is Euclidean PCA in the space of Cholesky coefficients of the covariance matrix. How would this produce different results compared to Euclidean PCA with respect to the SPD cone coordinates, or the proposed method?**
>
> We could indeed perform PCA using any other metric on SPD matrices, such as the Euclidean-Cholesky metric or the log-Euclidean metric. However, while we could visually compare the results, we have no apparent quantitative way to compare the results. Each method optimizes its own criterion, and any metric that one could think of to compare the methods would rely on a choice of underlying metric on the space of SPD matrices. Comparison of PCA methods with two different metrics thus boils down to comparing the metrics themselves. **We have added Figure 13 to illustrate this and included this discussion in Appendix A.2.**
>
> **Experiments: non-Gaussian case is missing obvious baselines. For the point cloud data presented, the most obvious way I can think of to do PCA is take some kind of off-the-shelf model like PointNet, and perform PCA in its latent space, and present the results. While this is certainly a much heavier method, I think some kind of visual comparison needs to be included in the main body, as this is the single most obvious performance question that readers might have.**
>
> It is true that a natural idea to perform PCA on 3D point clouds is to embed the point clouds into a latent space before applying classical PCA. We have therefore added this baseline in the revised version of our paper (see Appendix A.2 for the detailed experiment).
>
> Although natural, we did not conduct this experiment previously because we considered that the PointNet autoencoder + PCA pipeline has several limitations. First, training a point-cloud autoencoder requires a large collection of distributions. In our case (100 distributions), we therefore need to rely on a pretrained autoencoder trained on a related dataset. Second, PCA on autoencoder embeddings depends heavily on the geometry learned by the encoder. This learned geometry is not guaranteed to align with the Wasserstein structure (or any theoretically sound geometry), and the recovered principal components may not reflect meaningful modes of variation (as observed in the experiments of Appendix A.2, Figure 17). Third, for a given autoencoder architecture, different random initializations will lead to different learned geometries and thus different PCA components, meaning that the same principal components cannot be recovered robustly.
>
> However, we acknowledge that this is the first approach that one might attempt when seeking to perform PCA. **For this reason, we have added this experiment to the appendix (Section A.2) of our paper and included the above discussion. We have also added a paragraph on “Baselines” in Section 5 of the paper.**
>
> **Experiments: no evaluation of downstream applications of PCA like outlier detection. The authors do not evaluate how good the principal components produced are for any downstream purpose, instead they are just plotted to show their properties. I would like to see at least some kind of quantitative evaluation. For instance, this could be considered for outlier detection, to compare how well linear vs. the proposed geodesic PCA performs. Non-linear PCA has been studied for this specific purpose before, see for instance "Quadric Hypersurface Intersection for Manifold Learning in Feature Space" by Pavutnitskiy et al. which includes full experiments on this. Note that I am not suggesting the authors implement this specific downstream application of PCA, instead I think it is important to have at least some kind of evaluation of this kind.**
>
> The primary goal of our paper was to propose an algorithm for computing exact GPCA, for which only one algorithm existed in the literature in the 1D setting. However, we agree that discussing practical use cases is relevant in the paper. **We have added the experiment you suggested (see Appendix A.3 of the revised version)** to illustrate how GPCA can also be useful for outlier detection. Additionally, the MNIST experiment corresponding to Figure 12 (Appendix A.1), which was included in the first version of the paper, aims to demonstrate how one can perform classification and clustering after applying GPCA.
> In the revised version, we have added a sentence in the conclusion mentioning the possible applications of GPCA.
>
> **The overwhelming majority of my concerns with this paper concern the comprehensiveness of evaluations and specifically comparisons with various baselines like PCA on PointNet latent space. Please address those comparisons, or argue why they are not appropriate.**
>
> We hope that we have addressed your concerns in the responses above.

---

### Official Review · Reviewer_oUMT · 2025-11-02

**Soundness:** 4
**Presentation:** 4
**Contribution:** 4
**Rating:** 10
**Confidence:** 3

**Summary:**

This paper introduces a method for exact PCA in the Wasserstein space of probability measures P_2(R^d) under the Wasserstein metric. Instead of performing tangent PCA, which has been an established linearization technique for performing PCA on curved spaces (most heavily used in Riemannian settings) or geodesic PCA (Huckemann et al. Sommer et al.) that have tackled this problem in general Riemannian settings, this paper proposes two methods, i) for geodesic PCA for Gaussian measures, and a ii) neural network approach for parameter optimization for absolutely continuous probability measures. The authors show experimental evaluations on point clouds and image color distributions (mostly qualitative).

**Strengths:**

The work shows a method to compute principal modes of variation in datasets of probability measures, specifically using the Wasserstein geometry. For Gaussian measures, the method leverages the Bures-Wasserstein geometry and lifts computations to the space of invertible matrices, providing exact geodesics as principal components.

This is a significant contribution over earlier methods which have used linearized Wasserstein distances (Wang et al. (2013) and Boissard et al. (2015)), have approximated Geodesic PCA in 2D (Cazelles et al. (2018)), and the approach by Seguy & Cuturi (2015), which relied on  generalized geodesics to approximate PCA.


For general absolutely continuous distributions, the reliance on neural network parameterizations of Wasserstein geodesics is clever.

For the case of centered Gaussians, the application Bures-Wasserstein geometry and the decomposition via invertible matrices is well-motivated and theoretically grounded.

**Weaknesses:**

The block alternating algorithm for Gaussian GPCA is not guaranteed to always converge to a unique minimum due to non-uniqueness in the problem geometry (the authors acknowledge this).


In the general case, one needs to verify the eigenvalues of the Hessian at each step during the Otto geodesic update. This may be computationally expensive.

While the neural network implementation facilitates computational tractability, the construction of geodesics needs further tuning and learning from large examples and may have scalability issues.

The comparisons and experimental results are all qualitative and thus the practical application of this method is in question. However, this is an extremely minor point, not important for the publication of the paper.

**Questions:**

For experimental results, the authors compare their method against TPCA. How will their method compare with Bigot et al. 2017 (for 1D case)?

---

> ### Author Response · Authors · 2025-11-22
>
> Dear Reviewer,
>
> Thank you very much for taking the time to read our paper and for your positive feedback. We are very happy to read that you enjoyed our work. We have addressed your suggestions and questions below.
>
> **In the general case, one needs to verify the eigenvalues of the Hessian at each step during the Otto geodesic update. This may be computationally expensive.**
>
> We agree that in high-dimensional settings, computing the full Hessian for each sample in the current batch can be costly. Therefore, one can leverage algorithms that avoid forming the full Hessian and instead rely on matrix-vector products by estimating only the smallest and largest eigenvalues of the Hessian, which is comparatively less expensive. One such method is the LOBPCG algorithm [R1]. We tested this method in our experiments but did not observe a speed-up, as the dimensions we work with are relatively small. However, this algorithm would most likely be efficient in higher-dimensional settings.
>
> Moreover, in high dimensions, the number of samples required to accurately evaluate the extremal eigenvalues grows rapidly with the dimension. In such cases, an alternative strategy could involve an adversarial approach that keeps track of the eigenvectors corresponding to the worst-case eigenvalues.
>
> [R1] Duersch, J.A., Shao, M., Yang, C., Gu, M.: A robust and efficient implementation of LOBPCG. SIAM J. Sci. Comput. 40(5), 655–676 (2018). https://doi.org/10.1137/17M1129830
>
> **While the neural network implementation facilitates computational tractability, the construction of geodesics needs further tuning and learning from large examples and may have scalability issues.**
>
> Indeed, when relying on neural networks, we need a sufficient number of samples from each $\nu_i$. For the architecture and optimization of $f_\psi$ and $\varphi_\theta$, we use very generic multi-layer perceptron architectures, optimized with the default parameters of Adam and a learning rate of $5\mathrm{e}{-4}$. Moreover, the same architectures and optimization parameters are applied across all experiments, which indicates that the optimization method is relatively robust. The regularization coefficients $\lambda_I$ and $\lambda_O$ also appear to be robust (see the newly added Tables 2 and 3 in Appendix E.2.1 and E.2.2), and setting them to $1.0$ worked well consistently across all our experiments.
>
> **The comparisons and experimental results are all qualitative and thus the practical application of this method is in question. However, this is an extremely minor point, not important for the publication of the paper.**
>
> We acknowledge that the focus of this work was to propose a new methodology for solving GPCA, and that we did not discuss potential practical use cases. We have added a sentence in the conclusion mentioning the possible applications of GPCA. These include classification and clustering following GPCA (see the illustrative experiment in Figure 12), as well as outlier detection (see the newly added Section A.3 in the Appendix).
>
> **For experimental results, the authors compare their method against TPCA. How will their method compare with Bigot et al. 2017 (for 1D case)?**
>
> In Bigot et al. (2017), the authors propose log-PCA in one dimension, which is tangent PCA in our terminology. In Cazelles et al. (2018), the authors solve the exact geodesic PCA problem for one dimensional distributions (thanks to a convex-constrained algorithm), which corresponds to our formulation (1) of PCA for probability measures. They also provide a comparison between GPCA and TPCA in the one-dimensional Wasserstein space, which enjoys a simpler geometry.

---

### Author Response · Authors · 2025-12-03

Dear AC, SAC and Reviewers,

We would like to once again thank all reviewers for their interest in our work and for the time they dedicated to reviewing our paper. We have used their comments from the discussion phase to further improve our draft (the changes appear in red in the latest version of the paper). We are also pleased that our answers were well received, as noted, for instance, by Reviewers 2t9m and Pw4V, who indicated that they raised their scores.

During this rebuttal period, we have made several changes to our paper to address their remarks:
- We added an experiment (Appendix E.2.1 and E.2.2 in the updated version) showing that the algorithm we propose for the general case of a.c. probability measures is robust across a broad range of regularization coefficients $\lambda_I$ and $\lambda_O$.
- We included a non-synthetic experiment for the Gaussian version of GPCA in Section 5.1, based on the Weather CORGIS dataset.
- We added another baseline for the general-case algorithm, namely PointNet autoencoder + PCA, along with a paragraph in Appendix A.2 explaining why, although it may seem natural, this baseline has several limitations.
- We have emphasized in the Discussion the interest of GPCA with respect to downstream tasks such as classification, clustering, and outlier detection, and illustrated this point with an additional experiment (Appendix A.3) using GPCA for outlier detection, following the suggestion of Reviewer 2t9m.

These changes further justify the interest of studying GPCA in Wasserstein space, both from a practical and theoretical point of view. We also used the reviewers’ feedback to correct and improve many other parts of the draft and appreciated their careful reading.

Best regards,

The authors

---

### Meta-Review · Area_Chair_czCw · 2025-12-31

**Summary:**

The authors propose Geodesic PCA (GPCA) for probability measures in Wasserstein geometric space. In particular, the authors provide an exact geometrically GPCA for centered Gaussian measures; and a neural network parameterization of Wasserstein geodesics for general absolutely continuous measures.

The Reviewers agree that the finding results are novel, the proposed approaches are technically grounded, which advances the state of the art beyond existing linearization-based approaches. For the revised manuscript in the rebuttal, the authors provide additional empirical results, e.g., additional baselines, real-world dataset, additional task, which significantly alleviate raised concerns from the Reviewers. Some limitations on the convergence guarantees, and challenging computation may still remain.

Overall, we think that the submission is novel with high-quality technical finding results, additional results in the rebuttal strengthen empirical supports. The finding results are broadly interesting for the optimal transport, geometric machine learning community for analysis on distributional data.

**Reviewer Concerns:**

The Reviewers have some following concerns:

+ Reviewer oUMT: non-uniqueness convergence of the block alternating algorithm for Gaussian GPCA; computationally expensive for the eigenvalues of the Hessian at each step during the Otto geodesic update; scalability challenge for neural network approach;

+ Reviewer 2t9m: effects of orthogonal regularization in neural network objectives; weak experiment settings for Gaussian case; empirical support for Gaussian case beyond synthetic evaluation?; baselines for non-Gaussian experimental settings; downstream tasks?

+ Reviewer F5cW: scalability; a lack of practical applicability

+ Reviewer Pw4V: convergence of the proposed algorithms; the role of Proposition 6, 7, 8 for the proposed algorithms?

**Reviewer Scores:**

In the rebuttal, the authors provide several additional empirical results, e.g., more baselines, real-world task, downstream task, which significantly alleviate raised concerns from the Reviewers. Overall, we think the proposed approach is novel; its technical finding results are sound; the additional empirical results help to strengthen empirical evidence to support the proposed approaches.

---

### Decision · Program_Chairs · 2026-01-26

Accept (Oral)